# Sulfate residuals on Ru catalysts switch $CO_2$ reduction from methanation to reverse water-gas shift reaction

Min Chen[1], Longgang Liu[2], Xueyan Chen[1], Xiaoxiao Qin[1], Jianghao Zhang[1], Shaohua Xie [3], Fudong Liu [3] ✉, Hong He [1,4] & Changbin Zhang [1,4] ✉

Efficient heterogeneous catalyst design primarily focuses on engineering the active sites or supports, often neglecting the impact of trace impurities on catalytic performance. Herein, we demonstrate that even trace amounts of sulfate ($SO_4^{2-}$) residuals on Ru/TiO$_2$ can totally change the $CO_2$ reduction from methanation to reverse-water gas shift (RWGS) reaction under atmospheric pressure. We reveal that air annealing causes the trace amount of $SO_4^{2-}$ to migrate from TiO$_2$ to Ru/TiO$_2$ interface, leading to the significant changes in product selectivity from $CH_4$ to CO. Detailed characterizations and DFT calculations show that the sulfate at Ru/TiO$_2$ interface notably enhances the H transfer from Ru particles to the TiO$_2$ support, weakening the CO intermediate activation on Ru particles and inhibiting the further hydrogenation of CO to $CH_4$. This discovery highlights the vital role of trace impurities in $CO_2$ hydrogenation reaction, and also provides broad implications for the design and development of more efficient and selective heterogeneous catalysts.

At present, the atmospheric $CO_2$ level has surged to a historically high of approximately 416 ppm, further reinforcing the existing concerns about its significant contribution to global climate change[1–4]. Long treated as waste, $CO_2$ is now considered as a potentially useful carbon source for producing fuels and chemicals through photocatalytic, electrocatalytic, and thermal catalytic reduction in the presence of H$_2$, which can be obtained from water splitting using solar, wind, or other renewable energy sources[5–8]. Thermal catalytic reduction typically offers high reaction efficiency and is extensively used in practical applications. By designing supported metal catalysts and controlling the reaction conditions, a wide range of products, such as methane ($CH_4$)[9–11], carbon monoxide (CO)[12,13], methanol ($CH_3OH$)[14,15], and even long-chain hydrocarbons can be obtained[16].

Catalytic hydrogenation of $CO_2$ at atmospheric pressure typically involves either methanation reaction, yielding $CH_4$, or reverse water-gas shift (RWGS) reaction, leading to CO production[1,2,16–19]. Both

methanation and RWGS reactions play crucial roles in industrial processes related to hydrogen utilization and synthesis gas production. $CO_2$ hydrogenation to $CH_4$ or CO with high selectivity is desired according to specific application requirements but remains challenging. Previous research has established that the supported Ru/TiO$_2$ catalysts are one of the most active and stable catalysts in $CO_2$ hydrogenation reaction[20–22]. Very recently, some reports have highlighted that the crystal structure of TiO$_2$ support significantly impacted the selectivity of $CO_2$ hydrogenation on Ru/TiO$_2$ catalysts[18,23–25]. Qiao at al. observed that the selectivity of $CO_2$ hydrogenation could be completely reversed when Ru particles were supported on anatase-TiO$_2$ (high CO selectivity) versus on rutile-TiO$_2$ (high $CH_4$ selectivity)[23]. This phenomenon was attributed to the different electron transfer processes from Ru to the TiO$_2$ supports as a result of varying extents of hydrogen spillover related to crystal structure[23]. Wang et al. reported that annealing Ru/rutile-TiO$_2$ in air enhanced the $CO_2$ conversion to

[1]State Key Joint Laboratory of Environment Simulation and Pollution Control, Research Center for Eco-Environmental Sciences, Chinese Academy of Sciences, Beijing, China. [2]School of Chemistry and Chemical Engineering, Qufu Normal University, Qufu, China. [3]Department of Chemical and Environmental Engineering, Bourns College of Engineering, Center for Environmental Research and Technology (CE-CERT), Materials Science and Engineering (MSE) Program, University of California, Riverside, CA, USA. [4]University of Chinese Academy of Sciences, Beijing, China. ✉e-mail: fudong.liu@ucr.edu; cbzhang@rcees.ac.cn

CH₄, while annealing Ru/anatase-TiO₂ in air decreased the CO₂ conversion and led to CO production. They ascribed these differences to the contrasting metal-support interaction between Ru and anatase or rutile[18]. In contrast, Debecker et al. found that Ru supported on anatase, rutile, or a mixture of the two exhibited a variety of CO₂ conversions, but high CH₄ selectivity was observed on all catalysts[24]. The disparity in the observed CO or CH₄ selectivity among researchers indicates that the crystal structure of TiO₂ supports may not be the sole determining factor for the catalytic performance of CO₂ hydrogenation.

Upon careful examination of the literatures mentioned above, we noticed that the TiO₂ supports used in these studies were usually obtained from commercial sources, with some samples possibly containing trace amounts of residual impurities. Some impurities might remarkably affect the catalytic performance of Ru catalysts in CO₂ hydrogenation; however, their distinct significance was often overlooked in the course of research, potentially leading to flawed conclusions. In this work, we observed that the typical Ru/TiO₂ catalysts using both anatase and rutile supports displayed excellent performance in CO₂ methanation reaction. Surprisingly, the Ru/TiO₂ catalysts containing trace amount of SO₄²⁻ residuals showed no activity in CO₂ methanation, but excellent activity in RWGS reaction. This unique phenomenon suggested that the SO₄²⁻ residuals on these Ru catalysts, rather than the crystal structure of the TiO₂ supports, plays the key role in determining the catalytic performance of CO₂ hydrogenation. Further investigation revealed that annealing the sulfate-containing Ru/TiO₂ in air induced the sulfate migration from the TiO₂ support to the Ru/TiO₂ interface. At the interface, the sulfate could strongly promote the transfer of hydrogen from Ru particles to the TiO₂ support. The enhanced hydrogen spillover weakened the activation of CO intermediates on Ru particles, leading to significantly higher selectivity for CO production. This work not only introduces a novel viewpoint for

elucidating the variation of observed CO or CH₄ selectivity of CO₂ hydrogenation on Ru/TiO₂ catalysts, but also leads to a fundamental guideline for new catalyst design including the careful control of impurity levels and exploiting their positive impacts.

## Results
### The effects of sulfate on the catalytic performance of Ru/TiO₂ for CO₂ hydrogenation
A set of Ru/TiO₂ catalysts (with anatase TiO₂ purchased from Aldrich or Aladdin) were prepared by a wet impregnation method, with the Ru loading of 5 wt.%. The catalytic reduction of CO₂ was conducted at atmospheric pressure within the temperature range from 200 to 450 °C, in a fixed-bed flow reactor with a gas mixture composed of CO₂ (10 vol.%), H₂ (40 vol.%), and N₂ balance. The gas weight hourly space velocity (WHSV) was approximately 48,000 mL·g⁻¹·h⁻¹. The CO₂ hydrogenation under these conditions typically yields CH₄ via CO₂ methanation reaction and CO via RWGS reaction. Moreover, CO₂ methanation is more thermodynamically favorable compared to the RWGS reaction when the reaction temperature is below 500 °C[26]. Our calculation results about the thermodynamic equilibrium of CO₂ hydrogenation also showed that CH₄ was the favored product at lower temperature (<500 °C), while CO was the favored product at higher temperature (>500 °C) (Supplementary Figs. 1, 2). Thus, it remains an ongoing challenge to tune the high CH₄ selectivity to high CO selectivity at lower temperatures.

Figure 1a illustrates the products comparison on different sets of Ru/TiO₂ catalysts (the detailed comparison is provided in Supplementary Fig. 3). CH₄ was the main product on some Ru/TiO₂ catalysts, while CO was the main product on other Ru/TiO₂ catalysts. Considering that the purchased anatase TiO₂ supports may contain trace amount of sulfate species as impurity, we conducted the element analysis by inductively coupled plasma mass spectrometry (ICP-MS,

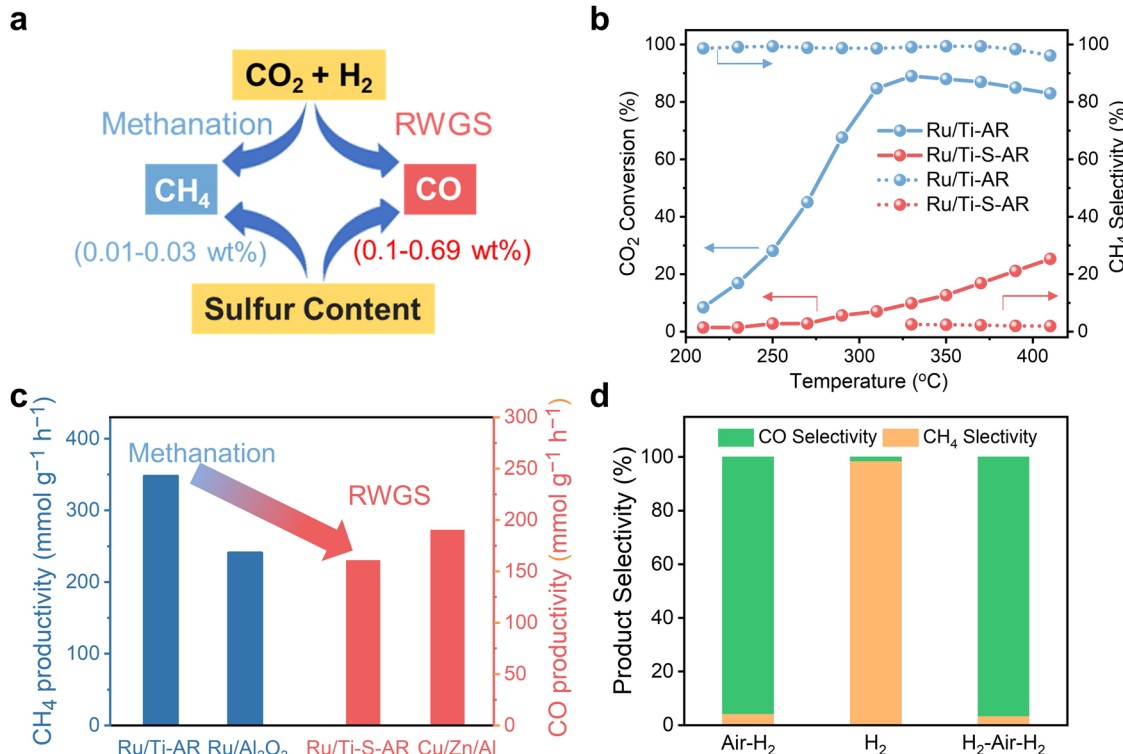

**Fig. 1 | Catalytic performance of the Ru/TiO₂ catalysts. a** The products and sulfur content comparison on the different sets of Ru/TiO₂ catalysts for CO₂ hydrogenation. **b** Temperature-dependent CO₂ conversion and CH₄ selectivity of Ru/TiO₂ catalysts with or without SO₄²⁻ species. **c** Comparison with commercial Ru/Al₂O₃ catalyst for CH₄ productivity at 350 °C and commercial CuO/ZnO/Al₂O₃ catalyst for CO productivity at 410 °C. **d** The product selectivity on Ru/Ti-S catalyst with air and/or H₂ pretreatment at 350 °C.

see Supplementary Table 1). The results revealed that these $TiO_2$ supports could be divided into two groups, with one group showing nearly no presence of $SO_4^{2-}$ (i.e., very low S content of 0.01–0.03 wt.%) and the other group showing the presence of trace amount of $SO_4^{2-}$ with relatively higher S content of 0.1–0.7 wt.%. Surprisingly, it was observed that the Ru/$TiO_2$ catalysts with no $SO_4^{2-}$ displayed high $CH_4$ selectivity, while the Ru/$TiO_2$ catalysts with trace amount of $SO_4^{2-}$ displayed high CO selectivity. These results strongly indicate that the presence of trace amount of $SO_4^{2-}$ on Ru/$TiO_2$ may play a crucial role in impacting the selectivity of $CO_2$ hydrogenation reaction.

To further investigate the influence of $SO_4^{2-}$ on the catalytic performance of $CO_2$ hydrogenation, we prepared the sulfate-free Ru/$TiO_2$ catalysts, in which the sulfate-free $TiO_2$ were synthesized by hydrolyzing tetrabutyl titanate, and also prepared the Ru/$TiO_2$ catalysts containing sulfate by purposely adding ammonium sulfate during the preparation process (with mole ratio of S/Ru set as 0, 0.03, 0.05, and 0.1). Before testing and characterization, the obtained sulfate-free and sulfated Ru/$TiO_2$ catalysts were annealed in air at 400 °C and then reduced by $H_2$ at 500 °C (denoted as Ru/Ti-AR and Ru/Ti-S-AR, Ru/Ti = Ru supported on anatase $TiO_2$, S = sulfated, AR = air annealing and $H_2$ reduction). Figure 1b shows that the Ru/Ti-AR exhibited excellent activity for $CO_2$ hydrogenation between 150 and 410 °C, and the $CO_2$ conversion reached the highest of *ca.* 89% at 330 °C and slightly decreased to 83% at 410 °C. The $CH_4$ selectivity maintained above 95% within this temperature range. With increasing the mole ratio of S/Ru from 0 to 0.1 (note that the ratio of S/Ru in Ru/Ti-S-AR was 0.1), the $CO_2$ conversion dropped sharply and the product distribution dramatically changed from $CH_4$ to CO (Supplementary Fig. 4). Besides, we tested the activity of the Ru/Ti-S-AR by altering contact time (Supplementary Fig. 5). The results showed that the $CO_2$ conversion was enhanced by increasing contact time, but high CO selectivity still remained, further confirming the high CO selectivity on Ru/Ti-S-AR. Compared with the commercial Ru/$Al_2O_3$ catalyst for $CO_2$ methanation and CuO/ZnO/$Al_2O_3$ catalyst for RWGS reaction (Fig. 1c), the $CH_4$ production on Ru/Ti-AR at 350 °C was 348 mmol g$^{-1}$ h$^{-1}$, which was 1.4 times higher than that on Ru/$Al_2O_3$. The CO production on Ru/Ti-S-AR at 410 °C was 160 mmol g$^{-1}$ h$^{-1}$, which was close to that on Cu/Zn/Al (191 mmol g$^{-1}$ h$^{-1}$). These results suggest that the Ru/Ti-AR performed as an efficient catalyst for $CO_2$ methanation reaction, while the Ru/Ti-S-AR performed as an efficient catalyst for RWGS reaction.

Considering that the crystal structure of $TiO_2$ was frequently discussed in influencing the product selectivity on Ru/$TiO_2$ catalyst in $CO_2$ hydrogenation reaction, we also synthesized a series of Ru/rutile catalysts, including the sulfate-free Ru/rutile-AR and Ru/rutile-R, as well as the sulfate-containing Ru/rutile-S-AR and Ru/rutile-S-R (AR = air annealing and $H_2$ reduction, S = sulfated, R = direct $H_2$ reduction). Upon analyzing the testing results depicted in Supplementary Fig. 6, we observed the similar trends as that on Ru/Ti catalysts. In short summary, both the sulfate-free Ru/$TiO_2$ catalysts (using anatase or rutile as support) showed high $CH_4$ selectivity, while the sulfate-modified Ru/$TiO_2$ showed high CO selectivity. These results emphasized that the presence of sulfate residuals in $TiO_2$ support, rather than the crystal structure of $TiO_2$, was the key factor influencing the catalytic performance of $CO_2$ hydrogenation.

Pretreating catalysts under different atmospheres commonly impacted the performance of catalysts in many reactions. Figure 1d shows that the high CO selectivity on Ru/Ti-S-AR was observed at 350 °C by annealing the as-prepared Ru/Ti-S in air followed by $H_2$ reduction. The pretreatment condition was also switched to direct $H_2$ reduction without pre-annealing in air. Surprisingly, the high $CH_4$ selectivity was observed (Fig. 1d). Afterwards, this sample was further treated with air annealing and subsequent $H_2$ reduction, and the product selectivity could achieve high CO (see detailed activity results as shown in Supplementary Fig. 7). In contrast, when the as-prepared sulfate-free Ru/Ti was pretreated under similar conditions, no such switch of product selectivity was observed at all (Supplementary

Fig. 8). These findings suggest that when there was trace amount of sulfate species on Ru/$TiO_2$ catalysts, annealing the catalysts in air was highly crucial for regulating the product selectivity in $CO_2$ hydrogenation.

## Sulfate-induced structural modification of Ru/$TiO_2$

To understand the effect of sulfates on the structure of Ru/$TiO_2$ catalysts, we investigated the geometric states of Ru nanoparticles (NPs) on different samples. The Ru/$TiO_2$ catalysts were prepared using the traditional wet impregnation method, which typically resulted in a wide range of metal dispersion on the support. High-angle annular dark-field scanning transmission electron microscopy (HAADF-STEM) measurement of the Ru/Ti-S-AR catalyst revealed numerous Ru particles distributed on the $TiO_2$ support, with sizes ranging from 5 to 12 nm (average size of 5.7 nm) (Fig. 2a and Supplementary Fig. 9). The Ru/Ti-AR exhibited a similar size distribution of Ru particles to that of Ru/Ti-S-AR. As shown in Supplementary Fig. 10, the Ru size distribution on Ru/Ti-R was in the range of 1–7 nm, with an average size of 2.9 nm, which was also comparable to that on Ru/Ti-S-R. The sizes of Ru derived from HAADF-STEM were in good agreement with XRD data (Supplementary Fig. 12). A summary about the Ru particles size and dispersion was shown in Supplementary Table 2. These results suggested that the air annealing at high temperature followed by $H_2$ reduction led to a higher degree of Ru particle aggregation compared with direct $H_2$ reduction treatment, and the presence of sulfate had negligible influence on the Ru particle size distribution.

The distribution of sulfate species on Ru/Ti-S-AR and Ru/Ti-S-R was investigated using energy dispersive X-ray (EDX) mapping. As shown in Fig. 2b, d, the S element on Ru/Ti-S-AR tended to accumulate near Ru particles, while the sulfates on Ru/Ti-S-R were randomly distributed on Ru/Ti-S-R. The distribution pattern of Ru and S elements on Ru/Ti-S-R-AR was similar to that on Ru/Ti-S-AR (Supplementary Fig. 11). This distinct relationship between Ru and S element distribution suggested that the air annealing could effectively drive the migration of sulfates on $TiO_2$ to the Ru/$TiO_2$ interface. In general, the sulfate species are bonded with $TiO_2$ through the chemical bonding of Ti-S. However, when the Ru species are loaded on $TiO_2$, a stronger chemical bonding of Ru-S may be present. During the air annealing at high temperatures, the surface sulfates on $TiO_2$ likely migrated to the Ru/$TiO_2$ interface to form a more stable state with the stronger chemical bonding of Ru-S (as illustrated in Fig. 2e).

The spatial distribution of sulfates differed noticeably between Ru/Ti-S-R and Ru/Ti-S-AR, implying that the chemical states of sulfate species were also expected to be distinct. To prove this point-of-view, we performed X-ray photoelectron spectroscopy (XPS) measurement. The S $2p_{1/2}$ spectra between binding energies of 160–172 eV are shown in Fig. 3a. For both Ru/Ti-S-AR and Ru/Ti-S-R, the doublet peaks of S $2p_{1/2}$ were observed at 169.4 and 168.2 eV, corresponding to the presence of sulfate ions ($SO_4^{2-}$)[27]. In addition, for Ru/Ti-S-AR, the peaks at 161.5 and 162.5 eV were detected, indicating the presence of $S^{2-}$[28], while these two peaks were considerably weaker for Ru/Ti-S-R. The presence of $SO_4^{2-}$ could be attributed to the introduction of $(NH_4)_2SO_4$ during the preparation process. The $SO_4^{2-}$ bonded on the surface of $TiO_2$, while the appearance of $S^{2-}$ was indicative of the formation of Ru-S bonds, confirming that the air annealing indeed facilitated the migration of a certain amount of sulfate from the $TiO_2$ support to the Ru/$TiO_2$ interface.

$H_2$-temperature-programmed reduction ($H_2$-TPR) experiments were performed to investigate the reducibility of catalysts and the interactions at the metal-support interfaces. The $H_2$-TPR results (Fig. 3b) showed that all the samples exhibited two main reduction peaks, centered at 95–103 °C and 128–131 °C, corresponding to the reduction of surface $RuO_x$ species weakly and strongly interacting with $TiO_2$, respectively[25,29,30]. The surface $RuO_x$ species were typically reduced by $H_2$ easily, displaying peaks at lower temperatures (95–103 °C), whereas the interfacial $RuO_x$ required higher

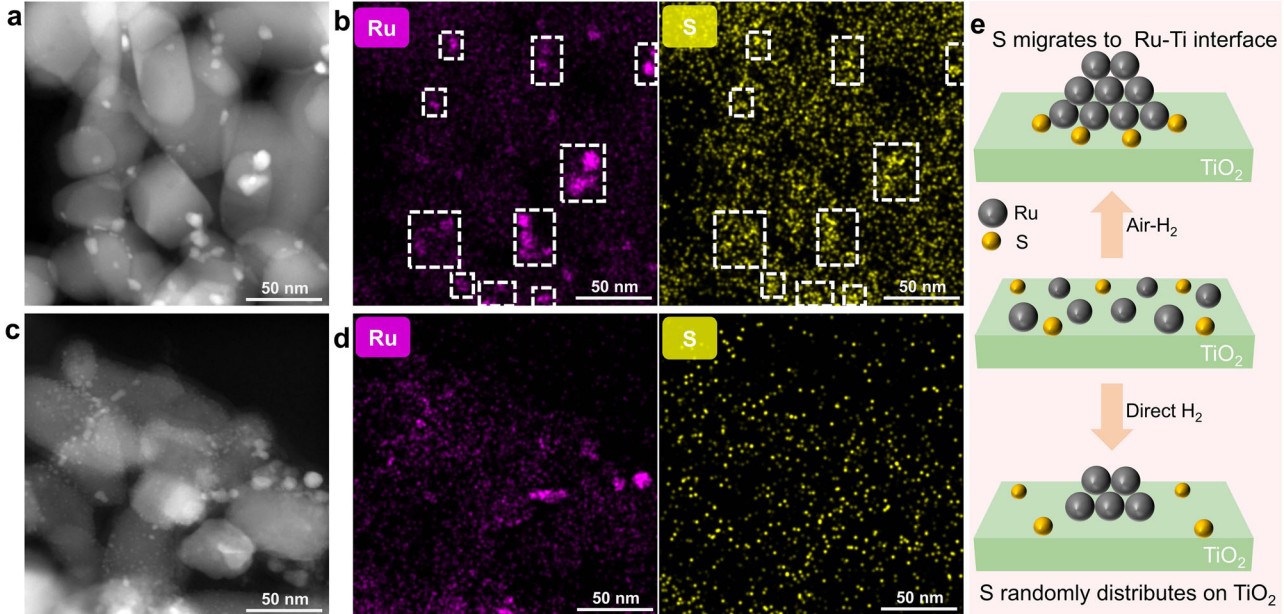

**Fig. 2 | The geometric states of Ru NPs on TiO₂. a** HAADF-STEM image of Ru/Ti-S-AR. **b** EDX mapping images of Ru and S elements on Ru/Ti-S-AR. **c** HAADF-STEM image of Ru/Ti-S-R. **d** EDX mapping images of Ru and S elements on Ru/Ti-S-R. **e** Schematic illustration of the evolution of Ru and S species on TiO₂ during air-H₂ or direct H₂ treatment.

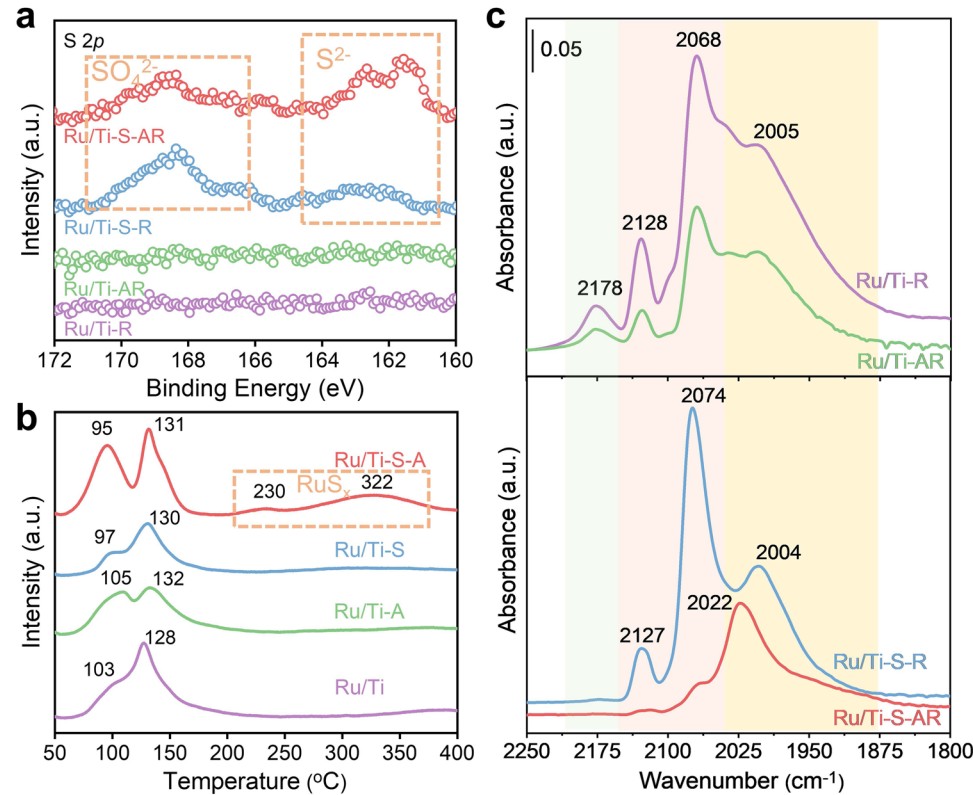

**Fig. 3 | Characterization on the chemical states of S and Ru species in different catalysts. a** S 2*p* XPS for Ru/Ti-S-AR, Ru/Ti-S-R, Ru/Ti-AR and Ru/Ti-R. **b** H₂-TPR profiles of Ru/Ti, Ru/Ti-A, Ru/Ti-S and Ru/Ti-S-A. **c** CO-DRIFTS on different catalysts at 25 °C, probing the surface states of Ru nanoparticles.

temperatures and exhibited H₂ reduction peaks at 128–131 °C. Furthermore, the ratio of interfacial $RuO_x$ to surface $RuO_x$ species in the Ru/Ti-S-R and Ru/Ti-R samples was noticeably higher than in the Ru/Ti-S-AR and Ru/Ti-AR samples. This observation could be attributed to the size difference of the Ru particles. The HAADF-STEM results

showed that the Ru particles in the Ru/Ti-S-R and Ru/Ti-R samples were smaller compared to those in the Ru/Ti-S-AR and Ru/Ti-AR samples. Smaller Ru particles typically exhibited more interfacial $RuO_x$ species on TiO₂, explaining the more abundant interfacial $RuO_x$ in the Ru/Ti-S-R and Ru/Ti-R samples. Notably, the Ru/Ti-S-AR sample also displayed

two additional peaks at 230 and 322 °C, which were not observed on other samples. These peaks could be attributed to the reduction of $RuS_x$[30,31], formed due to the migration of sulfate to the $Ru/TiO_2$ interface during air annealing. This phenomenon resulted in a strong interaction between Ru and sulfate, leading to the presence of $RuS_x$ peaks in $H_2$-TPR profile for Ru/Ti-S-AR sample.

To further investigate how the sulfate species induced the structural modification of $Ru/TiO_2$ catalyst, the chemical states of Ru were also characterized. The XPS results of Ru 3*d* for both Ru/Ti-AR and Ru/Ti-S-AR (Supplementary Fig. 14) revealed that the Ru species in the samples with or without sulfate were both in metallic state. Specifically, the presence of sulfate induced a slight shift of the metallic Ru peak from 279.7 to 279.9 eV, which might be due to the formation of Ru-S bonds at the $Ru/TiO_2$ interface[32]. For the impact of sulfate species on the structure $TiO_2$, no obvious changes were observed in Ti 2*p* on Ru/Ti-AR and Ru/Ti-S-AR (Supplementary Fig. 15), indicating that the presence of trace amount of sulfate species had negligible influence on the $TiO_2$ support, which was in line with the XRD and Raman results (Supplementary Figs. 12, 13).

To gain further insights into the surface states of Ru nanoparticles, we measured the CO adsorption at 25 °C using in situ diffuse-reflectance infrared Fourier transform spectroscopy (in situ DRIFTS). We firstly compared the CO adsorption on the sulfate-free Ru/Ti-R and Ru/Ti-AR samples. As shown in Fig. 3c, three CO vibrational bands at 2128, 2068, and 2005 $cm^{-1}$ appeared after CO adsorption on the sulfate-free samples, corresponding to the adsorption of CO on different Ru sites. Specifically, the bands at 2128 and 2068 $cm^{-1}$ could be assigned to the vibrations of CO on adsorbed on interfacial sites of Ru particles that interacted with the $TiO_2$[20,33], while the band at 2005 $cm^{-1}$ could be ascribed to the characteristic of CO adsorbed CO on top sites of Ru particles that interacted with all surrounding sites by Ru-Ru bonds[18,33]. In addition, the bands at 2178 $cm^{-1}$ could be ascribed to the CO adsorption on cationic Ti sites[34]. Notably, the intensities of the CO adsorption bands on Ru/Ti-AR were much lower than that on Ru/Ti-R, which could be due to the reduced exposure of Ru sites by air annealing. This observation was consistent with the HAADF-STEM results, which revealed that the Ru particle size in Ru/Ti-AR was apparently larger than that in Ru/Ti-R.

Next, the CO adsorptions on Ru/Ti-S-R and Ru/Ti-S-AR catalysts were examined. With the introduction of the sulfate to $TiO_2$, the CO adsorbed on $TiO_2$ (2178 $cm^{-1}$) nearly disappeared on Ru/Ti-S-R and Ru/Ti-S-AR, which should be due to the covering of cationic Ti sites by sulfate. In addition, the intensities of CO adsorption band associated with top sites of Ru particles (2004 $cm^{-1}$) and interfacial sites of Ru particles (2127 and 2074 $cm^{-1}$) on the Ru/Ti-S-R were comparable with the Ru/Ti-R, where only a very slight decrease was observed. This indicated that, during direct $H_2$ treatment, the introduced sulfate mainly stayed on $TiO_2$ and did not migrate to Ru particles. Notably, compared with the Ru/Ti-AR, the intensity of CO adsorption band (2127 and 2074 $cm^{-1}$) was dramatically decreased on the Ru/Ti-S-AR, implying that most interfacial sites of Ru particles were covered by sulfate. Meanwhile, CO adsorbed on top sites of Ru particles exhibited a shift to higher wavenumbers (from 2004 to 2022 $cm^{-1}$). This shift suggested an evolution of the top Ru atoms, in which the nearby interfacial Ru atoms occupied by sulfate might impact the Ru-CO bond of CO on top sites of Ru particles. The above results clearly showed that the sulfate on $TiO_2$ tended to migrate to Ru particles after air-$H_2$ treatment (Ru/Ti-S-AR), and did not migrate to Ru particles after direct $H_2$ treatment (Ru/Ti-S-R), which were consistent with the EDX mapping results.

## Origin of the catalytic performance modification by the introduction of trace sulfates

It was usually considered that the activity and selectivity of $CO_2$ hydrogenation on supported Ru catalysts were affected by the size of Ru particles. Single Ru sites or small Ru clusters less than 1 nm were suggested to be selective for $CO_2$ hydrogenation to CO, while larger Ru particles were typically more active for methanation reaction[21,22]. Our HAADF-STEM results indicated that the presence of surface sulfate had little influence on the size distribution of Ru particles. Moreover, the valence state distribution of Ru species was not obviously affected either by the surface sulfate species (Supplementary Fig. 14). Therefore, the effects of Ru particle size and the Ru valence state related to sulfates on the catalytic performance could be excluded. The CO-DRIFTS data strongly suggested that the sulfate species significantly modified the interfacial Ru sites, which were likely the main active sites controlling the $CO_2$ conversion and product selectivity. It was reported that the activation of $H_2$ and the transfer of H were critical steps in $CO_2$ hydrogenation[23]. Our $H_2$-TPR results demonstrated that the $H_2$ activation on Ru particles occurred easily at low temperatures (<150 °C), indicating that the $H_2$ activation was not the rate-determining step. Recent reports indicated that the strong hydrogen spillover, associated with enhanced H and electron migration, could lead to the reduced activation of intermediate CO, subsequently resulting in the distinctly low $CH_4$ selectivity[18,23]. This suggested that the sulfate species modifying the interfacial Ru sites might significantly influence the migration of H and electrons, thus affecting the catalytic performance of $Ru/TiO_2$ catalyst for $CO_2$ hydrogenation.

We conducted a comprehensive in situ DRIFTS study of $H_2$ reactions with different samples to investigate the migration of H atoms and electrons. In this process, $H_2$ molecules dissociated into H atoms at metallic sites and spilled over to O sites on the surface of $TiO_2$ forming localized Ti-O(H)-Ti species. Simultaneously, the electrons were donated into the shallow trap states in the band gap of $TiO_2$, leading to a broad IR absorbance in the spectrum[23,35–37]. As shown in Fig. 4a, the introduction of $H_2$ at 523 K resulted in a very broad absorbance across the range of 4000 to 1000 $cm^{-1}$ on Ru/Ti-S-AR, indicating the accumulation of electrons on the $TiO_2$ surface due to $H_2$ reduction. This broad IR absorbance was also observed on $Au/TiO_2$ and $Rh/TiO_2$, and it was attributed to the strong hydrogen spillover process[38,39]. In clear contrast, the Ru/Ti-R, Ru/Ti-AR, and Ru/Ti-S-R samples did not show such an adsorption feature, indicating that the intensity of hydrogen spillover on these samples was considerably lower than that on the Ru/Ti-S-AR. The temperature-dependent hydrogen spillover on Ru/Ti-S-AR was also depicted in Fig. 4b. As the temperature increased, the intensity of the broad absorbance across the range of 4000 to 1000 $cm^{-1}$ was significantly enhanced, signifying that the hydrogen spillover was facilitated at higher temperatures. These findings demonstrated that the presence of sulfate species, which modified the interfacial Ru sites, greatly enhanced the hydrogen spillover process.

We next carried out DFT calculations to investigate how the sulfate modification induced changes in the H migration. Based on the experimental results presented earlier, the theoretical calculations primarily focused on H migration reactions occurring at perimeter sites between $TiO_2$ and Ru (Supplementary Figs. 16 and 17). The calculations revealed that the H migration from Ru to the bridge O of $TiO_2$ on the sulfate-free $Ru-TiO_2$ interface had a barrier of 1.91 eV and an endothermicity of 0.09 eV. In contrast, when the $Ru/TiO_2$ interface was modified by sulfate, this barrier decreased to 1.47 eV, and the reaction became exothermic with an energy release of 0.46 eV (Fig. 4c). This result suggested that the sulfate modification greatly enhanced the H transfer process at the $Ru/TiO_2$ interface.

During the $CO_2$ hydrogenation on $Ru/TiO_2$ catalysts, the general reaction mechanism involved the initial adsorption of $CO_2$ at the Ru-$TiO_2$ interface, accompanied by $H_2$ activation and dissociation to H on the Ru sites[9,11,20,40,41]. With the assistance of dissociated H, the adsorbed $CO_2$ could be activated to form intermediate CO[11,20]. The presence of sufficient H and electrons allowed the intermediate CO to further convert into $CH_4$. We performed DRIFTS studies under steady-state $CO_2$ hydrogenation conditions, and it was observed that $CO_2$ was easily

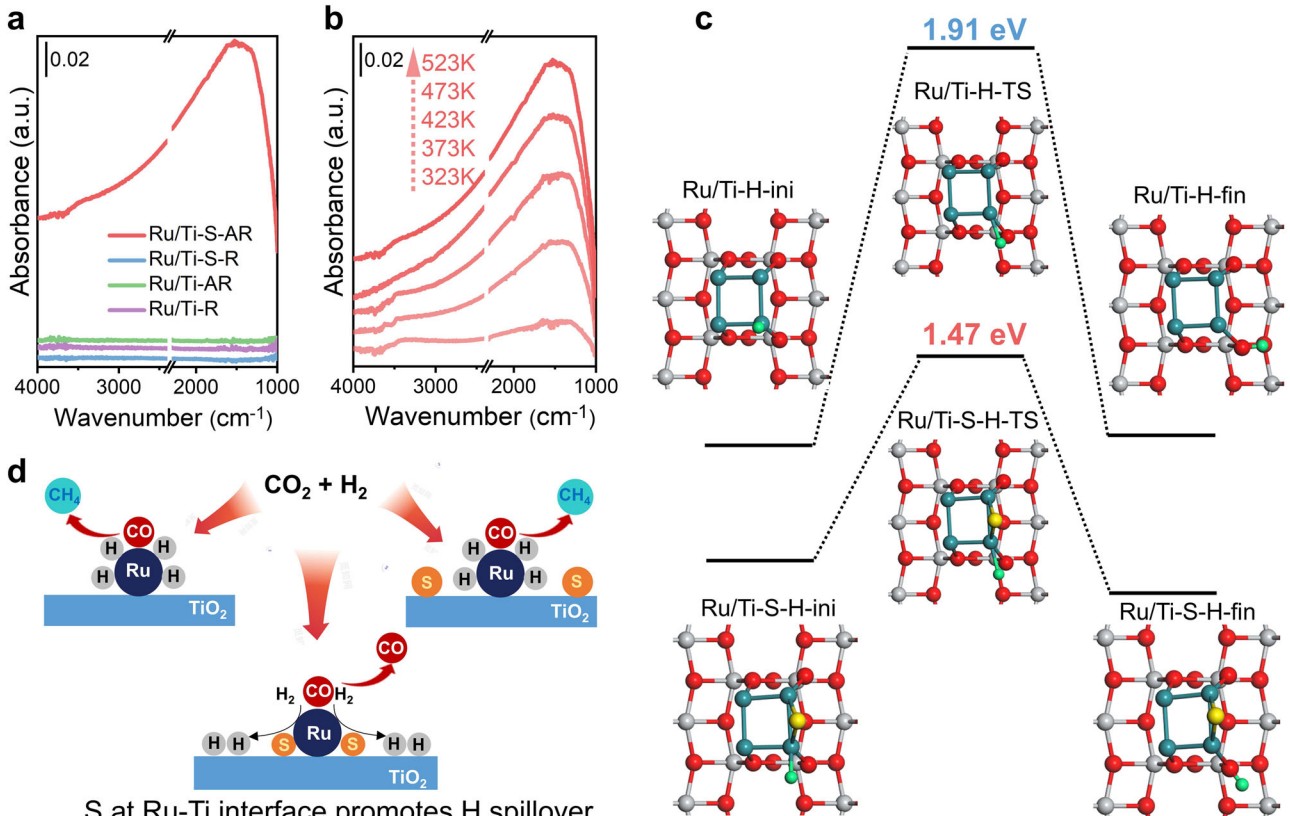

**Fig. 4 | Revealing the origin of catalytic performance modification by trace sulfates. a** In situ DRIFTS following the exposure to $H_2$ gas for Ru/Ti-S-AR, Ru/Ti-S-R, Ru/Ti-AR and Ru/Ti-R. **b** In situ DRIFTS following the exposure to $H_2$ gas for Ru/Ti-S-AR at different temperature. **c** DFT calculations of the H transfer process on sulfate-free and sulfate-containing Ru/TiO$_2$ catalysts (red, O; gray, Ti; cyan, Ru; yellow, S; green, H). **d** Schematic illustration of the mechanisms of $CO_2$ hydrogenation on sulfate-free and sulfate-containing Ru/TiO$_2$ catalysts.

converted to intermediate CO on Ru/Ti-AR and Ru/Ti-S-AR at low temperature. Additionally, the intermediate CO adsorbed at Ru site of Ru/Ti-AR could be converted to $CH_4$ when the reaction temperature was above 473 K, while the intermediate CO adsorbed at Ru site of Ru/Ti-S-AR was stable and no $CH_4$ was obtained (Supplementary Fig. 18). In the case of Ru/Ti-S-AR, where the Ru-TiO$_2$ interface was modified by sulfate, the H transfer process was greatly enhanced. This led to more H and electrons migrating from Ru to TiO$_2$ via the S medium, resulting in fewer H atoms remaining on the Ru sites. Consequently, this could effectively result in the low product selectivity to $CH_4$. In contrast, on the sulfate-free Ru/TiO$_2$, the hydrogen spillover and charge transfer could not proceed effectively. Therefore, the hydrogenation of adsorbed CO proceeded more smoothly, leading to the high $CH_4$ selectivity (Fig. 4d).

Catalytic performance can be remarkably affected, both advantageously and detrimentally, by the presence of trace impurities. However, the distinct significance of these trace impurities is often underestimated during the research process, which can potentially lead to erroneous conclusions. For example, sulfate, while capable of acting as a catalyst poison, can induce deactivation or reduced efficiency on some catalysts[42]. Conversely, on other types of catalysts, sulfate can play a positive role by enhancing metal dispersion or functioning as a promoter such as in photocatalytic water splitting for hydrogen production[43]. Other impurities such as chlorine, fluorine, or alkali cations can also yield similar positive or negative outcomes on catalytic performance[44–46]. In this work, the presence of trace amount of sulfate species on Ru/TiO$_2$ could significantly change the product distribution from high $CH_4$ selectivity to high CO selectivity. We also observed the similar product selectivity inversion by sulfate

modification on other methanation catalysts such as the Rh/TiO$_2$ and Ni/TiO$_2$ catalysts (Supplementary Figs. 19, 20). These results further suggested the ubiquitous role of residual sulfate in controlling the product selectivity in the $CO_2$ hydrogenation. This groundbreaking discovery serves as a poignant reminder of the paramount importance of comprehending the intricate interplay between impurities and catalyst structure in the endeavor to design catalysts that are not only more efficient but also exhibit heightened selectivity. Researchers must account for the origin of impurities, control the impurity levels, and develop strategies to mitigate their disadvantageous effects while harnessing their positive impacts on catalytic performance.

## Discussion
In summary, we discovered that the presence of residual sulfate species in commercial TiO$_2$ support, rather than the crystal structure of TiO$_2$, played a pivotal role in determining the product selectivity on the Ru/TiO$_2$ catalysts during $CO_2$ hydrogenation. Sulfate-free Ru/TiO$_2$ catalysts exhibited high $CH_4$ selectivity, whereas Ru/TiO$_2$ catalysts containing residual sulfate species displayed high CO selectivity. The annealing process in air at high temperatures induced the migration of sulfate on TiO$_2$ to the Ru/TiO$_2$ interface, where the interfacial sulfate species acted as an intermediate between the Ru sites and TiO$_2$ support, significantly promoting the H transfer from the former to the latter. The strong H spillover on Ru/TiO$_2$ catalysts containing residual sulfate species weakened the further activation of CO intermediates, resulting in low $CO_2$ conversion but very high selectivity to CO. These findings shed light on the role of trace impurities in heterogenous catalysis, and they can inform future research and development into ever more efficient and selective heterogeneous catalysts.

## Methods

### Syntheses of Ru/TiO₂ catalysts

Tetrabutyl titanate (TBOT, 99.5%) and ruthenium(III) nitrosyl nitrate were purchased from Aladdin. Commercial $TiO_2$ supports were purchased from Sigma-Aldrich and Aladdin. Ammonium sulfate was purchased from Beijing Innochem Science & Technology Co., LTD. $TiO_2$ were synthesized by hydrolyzing TBOT in a mixture of anhydrous ethanol and distilled water with a molar ratio of $n(TBOT)/n(C_2H_5OH)/n(H_2O) = 1:15:4$. Distilled water was dropped into the mixture of TBOT and anhydrous ethanol. The obtained precipitates were next dried and calcinated at 400 °C for 2 h. All Ru/$TiO_2$ catalysts with 5 wt.% Ru were prepared using the impregnation method. $TiO_2$ supports and a certain amount of $Ru(NO_3)_3$ were mixed in distilled water with stirring. The solution was evaporated at 60 °C under vacuum until dry. The resulting samples were dried at 100 °C overnight and then were calcined at 400 °C for 2 h in air or directly reduced with $H_2$ to prepare various Ru/$TiO_2$ catalysts. For the samples with sulfate addition, ammonium sulfate was incorporated during the impregnation process (with mole ratio of S/Ru set as 0, 0.03, 0.05, and 0.1).

### Characterization

$N_2$ adsorption-desorption isotherms for the catalysts were measured at 77 K on a Quantachrome instrument. To remove the effects of other adsorbed species, all samples were degassed at 300 °C for 6 h under vacuum before the tests.

X-ray diffraction (XRD) analyses were performed on a Bruker D8 Advance diffractometer, using Cu Kα radiation ($\lambda = 0.15406$ nm) at 40 mA and 40 kV in the range 5° <2θ < 90° with a step size of 0.02°. The phase compositions of the catalysts were identified by comparison of the patterns with the Power Diffraction Files (PDF). The elemental analysis was performed using an inductively coupled plasma mass spectrometer (ICP-MS, Agilent 7700 s) equipped with a concentric nebulizer and a cyclonic spray chamber. High-angle annular dark field scanning transmission electron microscopy and element mapping images were taken by a JEOL JEM-ARM 200 F, operating at 200 kV.

X-ray photoelectron spectra measurements were carried out on an AXIS Supra instrument, using a standard Al Kα X-ray source (150 W) and a pass energy of 40 eV. The binding energies (BE) of spectra were adjusted by carbon calibration (C 1s = 284.8 eV).

$H_2$-temperature-programmed reduction ($H_2$-TPR) measurements were conducted using a Micromeritics Chemisorb 2920 analyzer. The samples (~100 mg) were placed into a U-shaped quartz tube and pretreated in an Ar (30 mL·min⁻¹) atmosphere at 300 °C for 0.5 h. Then the samples were heated from 50 to 800 °C at a heating rate of 15 °C·min⁻¹ in a 10% $H_2$/Ar (50 mL·min⁻¹) flow. The effluent gas was passed through a cold trap to remove $H_2O$, and the signal was recorded by a thermal conductivity detector (TCD). For CO pulse chemisorption dispersion measurements, 70 mg of calcined catalyst was loaded into a U-shaped sample tube and reduced at 673 K for 1 h in 10% $H_2$/Ar. The catalyst was then flushed with He for 30 min. After cooling the sample to 323 K, pulse chemisorption measurements were performed with 10% CO/He while monitoring the effluent with a TCD.

In situ diffuse reflectance infrared Fourier transform spectroscopy (in situ DRIFTS) was performed on a Thermo Nicolet iS50 spectrometer equipped with a smart collector, and a liquid $N_2$-cooled MCT detector. The flow of the feed gas mixture was controlled using mass flow meters. All the spectra were measured with a resolution of 4 cm⁻¹ and an accumulation of 32 scans. A background spectrum was subtracted from each spectrum. CO adsorption experiments were carried out at 50 °C, and the mixture gas contained 500 ppm of CO and $N_2$ balance at a total flow rate of 100 mL/min.

Spin-polarized DFT calculations were carried out using the Vienna Ab initio Software Package (VASP)[47]. The ion-electron interactions are described using the projector-augmented wave (PAW) method and Perdew–Burke–Ernzerhof (PBE) generalized gradient approximation (GGA) functional[48,49]. The $TiO_2$ anatase (101) surface was modeled with a slab consisting of three O-Ti-O layers and a 15 Å vacuum gap. The bottom layers were fixed to their bulk structure, while only the top layer was allowed to relaxed. The Ru/$TiO_2$ model was constructed according to previous reported literature[50]. A planewave with a cut-off energy of 400 eV was employed. Γ-point calculations were performed for geometry optimization, with the convergence criteria for the energy and force were set to 10⁻⁵ eV and 0.02 eV/Å, respectively. The transition states for H transfer reaction were identified by relaxing the force below 0.05 eV/Å via the climbing image nudged-elastic band (CINEB) method[51].

### Catalytic tests

The evaluation of the $CO_2$ hydrogenation reaction was carried out in a quartz tube fixed-bed reactor under atmospheric pressure. The catalyst (50 mg) was loaded into the quartz tube and reduced with 10 vol.% $H_2$/$N_2$ (40 mL·min⁻¹) at 300 °C for 30 min prior to the catalytic performance evaluation. Then, the $CO_2$ hydrogenation reaction was performed at 250–550 °C under the reaction atmosphere of 4 vol.% $CO_2$, 16 vol.% $H_2$ and $N_2$ balance. The total flow rate was 40 mL·min⁻¹, and the gas hourly space velocity (GHSV) was 48,000 mL·h⁻¹·g_{cat}⁻¹. The outlet stream was analyzed by an online infrared gas analyzer. The experimental data variability for the activity tests was less than ±5%. $CO_2$ conversion in the activity test was defined as ([$CO_2$]_{inlet} − [$CO_2$]_{outlet}/[$CO_2$]_{inlet} × 100%, and the selectivity for $CH_4$ and CO were calculated as [$CH_4$]/([$CH_4$] + [CO]) × 100% and [CO]/([$CH_4$] + [CO]) × 100%, respectively.

### Reporting summary

Further information on research design is available in the Nature Portfolio Reporting Summary linked to this article.

## Data availability

Source data are provided with this paper.

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

## Acknowledgements

This work was supported by the National Natural Science Foundation of China (22025604, 22276204) and the National Key R&D Program of China (2023YFC3708401). F.L. acknowledges the Startup Fund from the University of California, Riverside.

## Author contributions

M.C. contributed to the central idea, performed the experiments, analyzed the data, and wrote the initial draft of the manuscript. L.L. performed the experiments. X.C., X.Q., and J.Z. contributed to data analysis. S.X. contributed to the manuscript revision. F.L. contributed to the refining of ideas, performing the analysis with constructive discussions, and manuscript revision. H.H. contributed to the project administration. C.Z. contributed to the funding acquisition, refining the ideas, carrying out additional analysis, and manuscript revision. All authors approved the final version of the manuscript.

## Competing interests

The authors declare no competing interests.
