## [Transparent Peer Review file · Nature Communications]

Sulfate residuals on Ru catalysts switch CO₂ reduction from methanation to reverse water-gas shift reaction

Corresponding Author: Professor Fudong Liu

Version 0:

Reviewer comments:

Reviewer #1

(Remarks to the Author)

This paper reports on the roles of sulfate in changing the selectivity of CO₂ hydrogenation from methane to CO, and claims the mechanism by promoting H transfer. In the experiment ammonia sulfate was used to modify the surface states of Ru/TiO₂ surface. Although the results show a way to tune the catalytic activity of CO₂ hydrogenation, some critical issues present in the manuscript make it unsuitable to be accepted in Nat Commun.

- 1) Fig 1b displays CO₂ conversions of 100% at temperatures higher than 325 °C. To my knowledge this is unusual, because the equilibrium conversion of CO₂ is about 90% above 300 °C, which further reduces with increasing temperatures. So the catalytic data are questionable. The authors need to double-check the experiment details.
- 2) The discussion of H transfer through DFT calculation is not persuasive. If the authors think it is H transfer that alters the selectivity, they need to reveal the kinetics of H spillover.
- 3) How the schemes of Fig 2e and 2f are concluded from the TEM and EDS mappings? The structure is not supported.

Reviewer #2

(Remarks to the Author)

Novelty and very interesting results about the effect of trace amount of sulfate residuals on the performance of Ru catalysts in the catalytic CO₂ reduction were presented in the manuscript titled "Sulfate residuals on Ru catalysts promote H transfer and switch CO₂ reduction from methanation to reverse water-gas shift reaction". In this work, sulfate residuals were found to switch CO₂ reduction from methanation to reverse water-gas shift reaction. These phenomena were attributed to enhanced H transfer from Ru particles to TiO₂ support by the sulfate at Ru-TiO₂ interface. This discovery highlights the vital role of trace impurities in CO₂ hydrogenation reaction, rendering its publication in Nature Communications after considering the following comments and making the corresponding revisions.

My comments:

- 1) The size of Ru particles was considered as an important factor in determining selectivity. Single Ru sites or small Ru clusters less than 1 nm were suggested to be selective for CO₂ hydrogenation to CO. Though HAADF-STEM results indicated that sulfate had little influence on the size distribution of Ru particles, the CO-DRIFTS (Fig. 3c) showed a totally different adsorption behavior when sulfate species was introduced. The bands at 2127 cm⁻¹ and 2074 cm⁻¹ assigned to CO adsorption on top Ru sites disappeared almost, indicating that sulfate could cover these Ru sites with the formation of a patch structure on Ru particles. The author should discuss the effect of this geometric effect towards selectivity change.
- 2) The author declare that the valence state distribution of Ru species was not obviously affected by surface sulfate species (line 262, page 15). However, Supplementary Fig. 11 indicates that the presence of sulfate caused a slight shift of the metallic Ru peak from 279.7 to 279.9 eV. The effect of valence state on the selectivity should be further discussed.
- 3) The author studied the effect of sulfate on supports in product selectivity on Ru catalysts. However, the changing trends in the product selectivity on other methanation metal catalysts such as Rh and Ni when sulfate was introduced should be studied so as to verify its universality to tune the selectivity for CO₂ hydrogenation by sulfate species.
- 4) The scale bar for DRIFT spectra in Figure 3c should be provided.

Reviewer #3

(Remarks to the Author)

Manuscript Number: NCOMMS-24-14147

Title: Sulfate residuals on Ru catalysts promote H transfer and switch CO₂ reduction from methanation to reverse water-gas shift reaction

In this manuscript, it was mainly discovered that air annealing at high temperature induced the migration of sulfates to the Ru-TiO₂ interface, resulting in the formation of RuS_x. The chemical bonding of Ru-S could facilitate electron transfer from Ru to S and enhance hydrogen spillover from Ru to TiO₂, thus weakening the adsorption and activation of CO intermediates and leading to low CO₂ conversion and high CO selectivity during CO₂ methanation. The manuscript is well-written, and the main conclusions that the authors highlight can be derived from the results. Before proceeding to publication, we recommend properly addressing the following points:

1. In Fig. 1a, CO was the main product on Ru/Ti-Aldrich-4 with 0.58% S content, contrasting with its high CH₄ selectivity in Supplementary Fig. 3b.
2. According to thermodynamic equilibrium (Supplementary Fig. 2), CO₂ conversion can not reach 100% above 320 °C in Fig. 1b.
3. What temperature are the data in Fig. 1d and Supplementary Fig. 7 shown at? Neither figure caption nor main text specified it.
4. In the case of sulfur-free Ru/rutile (Supplementary Fig. 5), air annealing prior to H₂ reduction did not have any influence on CO₂ conversion. How about in the case of sulfur-free Ru/anatase? Ru/Ti-AR and Ru/Ti-R do have different Ru particle size distributions, 5-12 nm and 1-7 nm, respectively. Did the difference in Ru particle sizes affect their CO₂ conversion and CO selectivity? A comparison of temperature-dependent CO₂ conversion/selectivity between Ru/Ti-AR and Ru/Ti-R should be presented.
5. Plots such as supplementary Fig. 4 (and some others) should be supplemented with data obtained at iso-conversion (or close to). It is too hasty to conclude on a selectivity inversion, when conversion levels are so different. For example, decreasing markedly the contact time for the CH₄-selective catalyst might result in much lower CO₂ conversion at high temperature and possibly not so high CH₄ selectivity. Similarly, increasing contact time with the CO-selective catalyst will increase conversion; it would be interesting to check if CO selectivity remains so high in such case or if it moves towards CH₄. Authors should back up their strong claims with such control experiments.
6. Fig. 1a is not very informative (also it contains symbols which are not explained in caption); it would be advisable to find a way to capture the essence of the Table reported in the supplementary, in a plot that show quantitative results. For example CH₄ yield and CO yield as a function of S content?
7. It is a weakness of the submission that Ru particles size (= an important parameter!) is only assessed through electron microscopy. What is the number of particles counted for each size distribution reported? Ideally, dispersion should be discussed in quantitative terms, using chemisorption data, to prove beyond reasonable doubt that indeed Ru dispersion (related to nanoparticles size) is indeed similar with or without S contamination.
8. Rutile-based catalysts are briefly mentioned (line 130) and claimed to behave like anatase-based ones (according to catalytic activity data). It is known, however, that Ru on rutile TiO₂ can have a peculiar behaviour during annealing in air, due to the epitaxial match between RuO₂ and rutile TiO₂. Did authors observe any kind of restructuration of the support (to be checked e.g. by TEM, N₂-physisorption, XRD) after annealing, and is this affected by the presence of S?
9. Line 219: The citation regarding the reduction of RuS_x is not accurate. Reference 30 investigated the effect of alkali additives (Li, Na, K, Cs) without the presence of sulfate species. The following references, which reported the reduction of amorphous RuS₂, may be helpful: <https://doi.org/10.1021/cs100053e> & <https://doi.org/10.1016/j.jcat.2008.08.015>
10. Line 231: a 0.2 eV shift in XPS is indeed a very slight shift. Please note that surface acidity (which can be affected by the presence of S) can cause different levels of C contamination, which itself can cause differential charging effects and issues with BE calibration (see <https://doi.org/10.1002/cphc.201300411> and <https://doi.org/10.1016/j.pmatsci.2019.100591>). It is advised to also try to apply a BE calibration on the Ti 2p peak, and to check if the 0.2 eV shift in the Ru 3d peak still holds.
11. Line 234: The authors stated that the presence of sulfate showed no influence on the chemical states of surface Ti species. It would be better to elucidate that the presence of sulfate at Ru-TiO₂ interface could enhance hydrogen spillover from Ru to TiO₂, thereby creating Ti³⁺ shallow traps, as later revealed by in-situ DRIFTS under H₂. However, XPS may not be sensitive enough to observe this change in chemical states of Ti species.
12. Line 241: Reference 20 only assigned the bands at 2128 and 2068 cm⁻¹ to vibrations of multicarbonyl species on undercoordinated Ru atoms and CO on-top of Ru atoms, respectively. The band at 2178 cm⁻¹ was not present in the CO-DRIFTS spectra of Ru/γ-Al₂O₃ in reference 20. Could it be possible that the band at 2178 cm⁻¹ was attributed to CO adsorption on Ti⁴⁺ sites (<https://doi.org/10.1038/s41467-022-32934-5>)? Reasonably, this band disappeared in the CO-DRIFTS spectra of sulfate-containing Ru/TiO₂ catalysts (Fig. 3c).
13. Line 331 mentions the case of Al₂O₃ supported Ru catalysts and claims similar trends were observed (as compared to the TiO₂-supported ones). In the case of Al₂O₃, one should rather talk about a complete deactivation of the catalyst, in the presence of S! In this case also, it would be advisable to show the data for the full temperature range, for Ru/Al (CH₄ selectivity appears to start to drop at 375°C...)

Version 1:

Reviewer comments:

Reviewer #1

(Remarks to the Author)

In the revised manuscript the authors have corrected their data to support their arguments of promoted H-spillover by sulfate to change CO₂ hydrogenation product from CH₄ to CO. This research basically reports a phenomenon that sulfate modification can alter the product selectivity, but the discussion about underlying structure-function relationship is not correct at all. Overall, their argument cannot be effectively supported by the data. Many claims made in the manuscript are actually assumptions and guesses rather than conclusions derived from solid data supports. I do not support further consideration for publication on Nature communications. Some critical comments are listed here.

1. The TiO₂ supports used in this work should be anatase, although no XRD and Raman spectra have been provided in the main text and supporting information. It is well acknowledged by catalysis community that anatase TiO₂ is the most typical oxide to display hydrogen spillover effect. I do believe sulfate can promote H-spillover enough to change the product reactivity. There should be other changes in the reaction kinetics. Although the authors have conducted DRIFTS and DFT calculations to support promoted H-spillover, these methods actually cannot provide solid supports for H-spillover. They should clarify the mechanisms by studying how CO₂ is activated and converted by the catalysts. Unfortunately, no characterization and discussion on the reaction mechanisms, so it is rude to ascribe the mechanisms to H-spillover without sufficient experimental supports.
2. Some claims in this paper are not correct or misleading. "The enhanced hydrogen spillover weakened the activation of CO intermediates, leading to significantly higher selectivity for CO production." H-spillover does not definitely change selectivity from CH₄ to CO, but sometimes can also enhance methanation by changing reaction pathways.
3. It is very puzzling to form RuS_x through annealing sulfate with Ru/TiO₂ in air. What is the reducing reagent to reduce sulfate? The species of RuS_x actually are not supported by any solid data.
4. "We conducted a comprehensive in situ DRIFTS study of H₂ reactions with different samples to investigate the migration of H atoms and electrons." This method is not right. In situ DRIFTS probe the activation and intermediates of CO₂ but not H migration. It is very unusual that the FTIR peaks are so broad. The measurement conditions should be not well optimized before collecting spectra.

In summary, I cannot recommend to accept this paper for publication because of the misleading and confusing claims and insufficient supports.

Reviewer #2

(Remarks to the Author)

The authors have addressed all my comments satisfactorily. Especially, the new supplementary experiments on other methanation metal catalysts such as Rh and Ni have verified its universality to tune the selectivity for CO₂ hydrogenation by sulfate species, providing broad implications for the design and development of more efficient and selective heterogeneous catalysts. Thus, I recommend its publication in Nature Communications.

Reviewer #3

(Remarks to the Author)

We stand by our comments that the paper is innovative, interesting, and potentially impactful.

We acknowledge the efforts made by the authors to respond to reviewers' comments, and in particular to solve some of the issues we had pointed out earlier.

Based on their response, however, we still have some reservations on some of our comments, and we add two new comments:

Comment 2: The authors stated in their response that the activity of the samples was "retested". However, the CO₂ conversion data at low temperatures appear exactly identical to those in the original manuscript; only the data at higher temperatures (> 330 °C) are lower, following the thermodynamic equilibrium CO₂ conversion. Are the data presented in this figure obtained from different experiments? (previous experiment for low T, and new experiment for high temperature?). Error bars could be added.

Comment 5: Increasing the contact time can be achieved by reducing the flow rate of the feed or increasing the amount of catalyst. Both result in a decrease in WHSV. In Supplementary Fig. 8, increasing the contact time (lower WHSV) leads to lower CO₂ conversion; this is not logical and may be a mistake. The authors should check the accuracy of the data.

Comment 6: Fig. 1a still contains the names of the different catalysts in front of colored symbols; but these are not used in the rest of the figure. Why showing a list of samples, if one does not discuss the differences between them. Only the dichotomy between high and low S content is useful here.

Comment 7: The characterization section should also include details on CO chemisorption measurement.

Comment 8: we were expecting an answer based on XRD results (to check re-structuration of the crystalline phases), not XPS.

New comments:

1. The characterization section provides details on XRD analyses; however, no XRD results are presented and discussed. Showing XRD patterns could be useful, providing direct or indirect evidence (e.g., crystalline phases of anatase or rutile TiO₂, any possible phase transformation/restructuring, and size of Ru species).
2. Was the H₂-TPR experiment performed without pre-reduction under H₂? The catalysts' name in H₂-TPR profiles might be misleading. Should they be changed to Ru/Ti, Ru/Ti-A, Ru/Ti-S and Ru/Ti-S-A?

Reviewer #4

(Remarks to the Author)

This revised paper can be accepted, because moderate modifications were made.

Version 2:

Reviewer comments:

Reviewer #1

(Remarks to the Author)

The authors have provided more data in the revised manuscript and addressed reviewers' issues. Although other reviewers have thought this paper can be accepted for publication, I still strongly insist to my initial suggestion that this paper is not acceptable for publication for the academic credits of the authors and Nature Communications. They are many questionable data and misleading claims in this submitted manuscript.

1. In the first submitted version CO₂ conversions are 100% at temperature higher than 300 °C. This is not owing to the standard curve at all, because standard curves can only affect the values of CO₂ conversion and CH₄/CO selectivity. This makes me conclude some catalytic results are fabricated but not really measured. The corresponding authors should double check the raw data.
2. The authors claim S on the surface promotes H transfer and selectivity change. In the experimental section, no descriptions about how TiO₂ supports were prepared from tetrabutyl titanate and how the catalysts were modified with S were given. It is very ridiculous the most important experimental details are not described at all.
3. The authors ascribe the selectivity change to H spillover promoted by S. This is just a guess. Anatase TiO₂ is the most typical oxide support to display H spillover and strong metal-support interaction (SMSI). For Ru/TiO₂-anatase, even without S, H spillover is still an indispensable process in CO₂ hydrogenation reaction. Instead, many researches have shown SMSI can stably occur for Ru/TiO₂-anatase in CO₂ hydrogenation reaction, and can change the product from CH₄ to CO. This phenomenon is not found in the data shown by this paper. This further leads me to doubt the data. I cannot agree with explanation about the selectivity change by H transfer. This will mislead future researches in this area.

Reviewer #3

(Remarks to the Author)

Authors have provided satisfactory responses to my questions and comments. The corresponding changes in the MS solve the issues I was pointing. I have also checked the additional data provided in response to other reviewers comments, and found them both useful and valuable. I recommend this paper for publication.

Version 3:

Reviewer comments:

Reviewer #1

(Remarks to the Author)

The authors have made commendable efforts to substantiate their claims and results, although the conclusions are not strongly supported by current data. This can be a very excellent study that presents significant findings in the field of heterogeneous catalysis for CO₂ hydrogenation if the mechanisms can be clearly revealed. To convincingly demonstrate that sulfur (S) influences selectivity by promoting hydrogen spillover rather than through a strong metal-support interaction (SMSI) effect, I recommend that the authors provide operando FTIR data in the main text rather than in the supporting information. Additionally, a detailed discussion on how sulfur alters the catalytic pathways would strengthen the manuscript (see *Angew. Chem. Int. Ed.* 2020, 19983; *Nat. Commun.* 2022, 327; *JACS* 2018, 11241; *Angew. Chem. Int. Ed.* 2020, 22763 for relevant insights). To further rule out the influence of SMSI, I suggest comparing the CO adsorption behavior under different conditions, as this would offer a clear distinction between the two mechanisms. The DFT results actually can not effectively support the mechanisms here.

Point-to-Point Responses to the Editor and Reviewers' Comments

Reviewer #1:

This paper reports on the roles of sulfate in changing the selectivity of CO₂ hydrogenation from methane to CO, and claims the mechanism by promoting H transfer. In the experiment ammonia sulfate was used to modify the surface states of Ru/TiO₂ surface. Although the results show a way to tune the catalytic activity of CO₂ hydrogenation, some critical issues present in the manuscript make it unsuitable to be accepted in Nat Commun.

1) Fig 1b displays CO₂ conversions of 100% at temperatures higher than 325 °C. To my knowledge this is unusual, because the equilibrium conversion of CO₂ is about 90% above 300 °C, which further reduces with increasing temperatures. So the catalytic data are questionable. The authors need to double-check the experiment details.

Reply: Thank you for your valuable comments. The thermodynamic equilibrium CO₂ conversion *vs.* temperature has been shown in Supplementary Fig. 2. As you mentioned, the CO₂ conversions at temperatures higher than 325 °C indeed could not reach 100%. By rechecking the raw data of the activity tests, we found that the previous standard curve of CO₂ concentration exhibited inadequate accuracy when CO₂ was at low concentrations. Based on your suggestion, we rebuilt the standard curve of CO₂ concentrations and retested the activity of the samples, and the results showed that the CO₂ conversion on Ru/-TiO₂-AR reached the highest of *ca.* 89% at 330 °C and then slightly decreased to 83 % at 410 °C. All related data have been updated in revised manuscript, accordingly. We highly appreciate your advice on this matter.

The revisions have been included in the updated manuscript in lines 116-119 on page 7 “Fig. 1b shows that the Ru/Ti-AR exhibited excellent activity for CO₂ hydrogenation between 150 and 410 °C, and the CO₂ conversion reached the highest of *ca.* 89% at 330 °C and slightly decreased to 83% at 410 °C. The CH₄ selectivity maintained above 95% within this temperature range.”

Supplementary Fig. 2 | Influence of temperature on the thermodynamic equilibrium of the

CO₂ hydrogenation reaction at 1 bar.

Fig. 1 b. Temperature-dependent CO₂ conversion and CH₄ selectivity of Ru/TiO₂ catalysts with or without SO₄²⁻ species.

2) The discussion of H transfer through DFT calculation is not persuasive. If the authors think it is H transfer that alters the selectivity, they need to reveal the kinetics of H spillover.

Reply: Thank you for your kind comments and suggestion. Previous studies have demonstrated that H transfer on Ru-TiO₂ could greatly affect the selectivity in CO₂ hydrogenation, and substantial H transfer from Ru particles to TiO₂ would inhibit the deep hydrogenation to CH₄ (*Angew. Chem. Int. Ed.* 2020, 59, 19983; *Nature. Com.* 2022, 13, 327). We here noted that trace amounts of sulfate at Ru-TiO₂ interface played a key role in inducing the selectivity switch from high CH₄ selectivity to high CO selectivity, suggesting that the sulfate species at Ru-TiO₂ interface might have significantly influenced the H transfer, and therefore altered the selectivity.

By H₂-DRIFTS measurements (Fig 4a, 4b), we observed that, with the introduction of H₂ at 523 K, a very broad absorbance across the range of 4000 to 1000 cm⁻¹ on Ru/Ti-S-AR appeared, indicating the accumulation of electrons on the TiO₂ surface due to H₂ reduction. In contrast, the Ru/Ti-R, Ru/Ti-AR, and Ru/Ti-S-R samples did not show such adsorption feature, indicating that the intensity of H₂ spillover was very weak on these samples. The temperature-dependent H₂ spillover on Ru/Ti-S-AR showed that, as the temperature increased, the intensity of the broad absorbance across the range of 4000 to 1000 cm⁻¹ was also significantly enhanced, signifying that the H₂ spillover was facilitated at higher temperatures. These results further suggested that the surface sulfate species, which modified the interfacial Ru sites, greatly enhanced the H₂ spillover process.

Based on the experimental results, we then carried out the DFT calculations, primarily focusing on H migration reactions occurring at perimeter sites between TiO₂ and Ru. The calculation results revealed that the H migration from Ru to the bridge O of TiO₂ on the sulfate-free Ru-TiO₂ interface had a barrier of 1.91 eV and an endothermicity of 0.09 eV. In clear

contrast, when the Ru-TiO₂ interface was modified by sulfate, this barrier decreased to 1.47 eV, and the reaction became exothermic with an energy release of 0.46 eV (Fig. 4c). These results confirmed that the sulfate modification greatly enhanced the H transfer process at the Ru-TiO₂ interface. Hence, we claimed that the sulfate modification on Ru catalysts promoted H transfer, inducing the selectivity switch from high CH₄ selectivity to high CO selectivity. We hope these explanation well answers your question.

3) How the schemes of Fig 2e and 2f are concluded from the TEM and EDS mappings? The structure is not supported.

Reply: Thank you for your question. The distribution of sulfate species on Ru/Ti-S-AR and Ru/Ti-S-R was investigated using EDX mapping. As shown in Fig. 2b and Fig. 2d, sulfur element tended to accumulate around Ru species on Ru/Ti-S-AR (Fig. 2b), while the sulfur element was only randomly distributed on Ru/Ti-S-R (Fig. 2d). This different distribution of sulfur element suggested their distinct migration behaviors during air-H₂ or direct H₂ treatment. Specifically, sulfate on TiO₂ tended to migrate to Ru particles under air-H₂ treatment (Ru/Ti-S-AR), while it randomly migrated on TiO₂ under direct H₂ treatment (Ru/Ti-S-R). Upon the above analysis, we have presented a new scheme in Fig 2e.

Fig. 2 | The geometric states of Ru NPs on TiO₂. **a** HAADF-STEM image of Ru/Ti-S-AR. **b** EDX mapping images of Ru and S elements on Ru/Ti-S-AR. **c** HAADF-STEM image of Ru-Ti-S-R. **d** EDX mapping images of Ru and S elements on Ru/Ti-S-R. **e** Schematic illustration of the evolution of Ru and S species on TiO₂ during air-H₂ or direct H₂ treatment.

Thank you again for your kind comments and suggestions, which helped us greatly improve the quality of this manuscript. We hope that this manuscript can be accepted for publication in *Nature Communications* now.

Reviewer #2:

Novelty and very interesting results about the effect of trace amount of sulfate residuals on the performance of Ru catalysts in the catalytic CO₂ reduction were presented in the manuscript titled "Sulfate residuals on Ru catalysts promote H transfer and switch CO₂ reduction from methanation to reverse water-gas shift reaction". In this work, sulfate residuals were found to switch CO₂ reduction from methanation to reverse water-gas shift reaction. These phenomena were attributed to enhanced H transfer from Ru particles to TiO₂ support by the sulfate at Ru-TiO₂ interface. This discovery highlights the vital role of trace impurities in CO₂ hydrogenation reaction, rendering its publication in Nature Communications after considering the following comments and making the corresponding revisions.

My comments:

1) The size of Ru particles was considered as an important factor in determining selectivity. Single Ru sites or small Ru clusters less than 1nm were suggested to be selective for CO₂ hydrogenation to CO. Though HAADF-STEM results indicated that sulfate had little influence on the size distribution of Ru particles, the CO-DRIFTS (Fig.3c) showed a totally different adsorption behavior when sulfate species was introduced. The bands at 2127 cm⁻¹ and 2074 cm⁻¹ assigned to CO adsorption on top Ru sites disappeared almost, indicating that sulfate could cover these Ru sites with the formation of a patch structure on Ru particles. The author should discuss the effect of this geometric effect towards selectivity change.

Reply: Thank you for your kind comments and suggestions. We carefully re-analyzed the CO-DRIFTS results, and re-assigned the CO vibrational bands under the guidance of related literature. The low frequency band between 2000 and 2050 cm⁻¹ was assigned to the linearly adsorbed CO on top sites of Ru particles with all surrounding sites of Ru-Ru bonds. The high frequency band (2128, 2068 cm⁻¹) were associated with CO adsorbed on interfacial Ru sites interacting with the TiO₂ (*J. Am. Chem. Soc.* 2013, 135, 6107-6121; *Angew. Chem. Int. Ed.* 2020, 59, 22763-22770; *J. Catal.* 2018, 367, 194-205). The bands at 2178 cm⁻¹ could be ascribed to the CO adsorption on cationic Ti sites (*Nat. Com.* 2022, 13, 5186; *Catal. Com.* 2007, 8, 1715-1718). Based on these assignments, we discussed the influence of sulfate on the structure evolution of Ru/TiO₂ as follows:

After introducing the sulfate to Ru/Ti-R and Ru/Ti-AR catalysts, the CO adsorbed on TiO₂ (2178 cm⁻¹) nearly disappeared on Ru/Ti-S-R and Ru/Ti-S-AR, which should be due to the covering of cationic Ti sites by sulfate. The intensities of CO adsorption bands associated with top sites of Ru particles (2004 cm⁻¹) and interfacial sites of Ru particles (2127 and 2074 cm⁻¹) on the Ru/Ti-S-R were similar to those on Ru/Ti-R only with a very slight decrease, indicating that, during the direct H₂ treatment, the introduction of sulfate mainly stayed on TiO₂ and did not migrate to Ru sites. Notably, the intensities of CO adsorption bands at 2127 and 2074 cm⁻¹ dramatically decreased on Ru/Ti-S-AR, implying that most interfacial sites of Ru particles were covered by sulfate. Meanwhile, the CO band related to the top of Ru sites exhibited a blue shift from 2004 to 2022 cm⁻¹, suggesting an evolution of the top Ru atoms, in which the nearby interfacial Ru atoms occupied by sulfate might impact the Ru-CO bond on top Ru sites. The above results clearly showed that the sulfate on TiO₂ tended to migrate to Ru particles under air-H₂ treatment (Ru/Ti-S-AR), but did not migrate to Ru particles under direct H₂ treatment

(Ru/Ti-S-R), which were consistent with the EDX mapping results.

By using H₂-DRIFTS and DFT calculations (Fig 4a, 4b), we demonstrated that the geometric effect of sulfate covering on interfacial Ru site greatly enhanced the hydrogen spillover process. The substantial H atoms could transfer from Ru particles to TiO₂, which would inhibit the deep hydrogenation to CH₄, therefore altering the selectivity in CO₂ hydrogenation.

Fig. 3 | Characterization on the chemical states of S and Ru species in different catalysts. a S 2p XPS for Ru/Ti-S-AR, Ru/Ti-S-R, Ru/Ti-AR and Ru/Ti-R. **b** H₂-TPR profiles. **c** CO-DRIFTS on different catalysts at 25 °C, probing the surface states of Ru nanoparticles.

We have added detailed discussions in the revised manuscript in lines 238-267 on pages 14-15 “To gain further insights into the surface states of Ru nanoparticles, we measured the CO adsorption at 25 °C using *in situ* diffuse-reflectance infrared Fourier transform spectroscopy (*in situ* DRIFTS). We firstly compared the CO adsorption on the sulfate-free Ru/Ti-R and Ru/Ti-AR samples. As shown in Fig. 3c, three CO vibrational bands at 2128, 2068, and 2005 cm⁻¹ appeared after CO adsorption on the sulfate-free samples, corresponding to the adsorption of CO on different Ru sites. Specifically, the bands at 2128 and 2068 cm⁻¹ could be assigned to the vibrations of CO on adsorbed on interfacial sites of Ru particles that interacted with the TiO₂^{20,33}, while the band at 2005 cm⁻¹ could be ascribed to the characteristic of CO adsorbed CO on top sites of Ru particles that interacted with all surrounding sites by Ru-Ru bonds^{18,33}. In addition, the bands at 2178 cm⁻¹ could be ascribed to the CO adsorption on cationic Ti sites³⁴. Notably, the intensities of the CO adsorption bands on Ru/Ti-AR were much lower than that on Ru/Ti-R, which could be due to the reduced exposure of Ru sites by air annealing. This observation was consistent with the HAADF-STEM results, which revealed that the Ru particle

size in Ru/Ti-AR was apparently larger than that in Ru/Ti-R.

Next, the CO adsorptions on Ru/Ti-S-R and Ru/Ti-S-AR catalysts were examined. With the introduction of the sulfate to TiO₂, the CO adsorbed on TiO₂ (2178 cm⁻¹) nearly disappeared on Ru/Ti-S-R and Ru/Ti-S-AR, which should be due to the covering of cationic Ti sites by sulfate. In addition, the intensities of CO adsorption band associated with top sites of Ru particles (2004 cm⁻¹) and interfacial sites of Ru particles (2127 and 2074 cm⁻¹) on the Ru/Ti-S-R were comparable with the Ru/Ti-R, where only a very slight decrease was observed. This indicated that, during direct H₂ treatment, the introduced sulfate mainly stayed on TiO₂ and did not migrate to Ru particles. Notably, compared with the Ru/Ti-AR, the intensity of CO adsorption band (2127 and 2074 cm⁻¹) was dramatically decreased on the Ru/Ti-S-AR, implying that most interfacial sites of Ru particles were covered by sulfate. Meanwhile, CO adsorbed on top sites of Ru particles exhibited a shift to higher wavenumbers (from 2004 to 2022 cm⁻¹). This shift suggested an evolution of the top Ru atoms, in which the nearby interfacial Ru atoms occupied by sulfate might impact the Ru-CO bond of CO on top sites of Ru particles. The above results clearly showed that, the sulfate on TiO₂ tended to migrate to Ru particles after air-H₂ treatment (Ru/Ti-S-AR), and did not migrate to Ru particles after direct H₂ treatment (Ru/Ti-S-R), which were consistent with the EDX mapping results.”

References

18. Zhou, J. *et al.* Interfacial compatibility critically controls Ru/TiO₂ metal-support interaction modes in CO₂ hydrogenation. *Nat. Commun.* 13, 327 (2022).
20. Chen, S. *et al.* Raising the CO_x methanation activity of a Ru/gamma-Al₂O₃ catalyst by activated modification of metal-support interactions. *Angew. Chem. Int. Ed.* 59, 22763-22770 (2020).
33. Yan, Y. *et al.* Ru/Al₂O₃ catalyzed CO₂ hydrogenation: Oxygen-exchange on metal-support interfaces. *J. Catal.* 367, 194-205 (2018).
34. Kots, P. *et al.* Electronic modulation of metal-support interactions improves polypropylene hydrogenolysis over ruthenium catalysts. *Nat. Commun.* 13, 5186 (2022).

2) The author declare that the valence state distribution of Ru species was not obviously affected by surface sulfate species (line 262, page 15). However, Supplementary Fig.11 indicates that the presence of sulfate caused a slight shift of the metallic Ru peak from 279.7 to 279.9 eV. The effect of valence state on the selectivity should be further discussed.

Reply: Thanks for your valuable comments. As shown in Fig. 11, the presence of surface sulfate indeed induced a slight shift of the Ru 3d peak from 279.7 to 279.9 eV. According to previous studies (*Nature Catal.* 2020, 3, 454-462; *ACS Catal.* 2019, 9, 11088-11103), Ru 3d peaks at 279.7 or 279.9 eV could both be ascribed to metallic Ru species.

We understand that you might have concern about the effect of slight change in valence state of metallic Ru species on the CO₂ hydrogenation selectivity. Previous study has found that Ru supported on TiO₂ (001) and TiO₂ (101) exhibited very slight difference in the valence state of Ru (0.4 eV shift in XPS), however, both Ru-TiO₂ (001) and Ru-TiO₂ (101) showed high CH₄ selectivity in CO₂ hydrogenation (*RSC Adv.*, 2014, 4, 10834). Hence, it was reasonable to

conclude that such a slight change in valence state of metallic Ru species in our work would not impact the selectivity. We hope this has well addressed your concern.

3) The author studied the effect of sulfate on supports in product selectivity on Ru catalysts. However, the changing trends in the product selectivity on other methanation metal catalysts such as Rh and Ni when sulfate was introduced should be studied so as to verify its universality to tune the selectivity for CO₂ hydrogenation by sulfate species.

Reply: Thanks for your kind comments. Following your suggestions, we prepared the Rh/Ti-AR, Rh/Ti-S-AR, Ni/Ti-AR and Ni/Ti-S-AR catalysts and tested their activity, and the results are presented in Supplementary Figs. 16 and 17. The similar sulfate effect on product selectivity was also present on Rh/TiO₂ and Ni/TiO₂ catalysts, where trace amounts of sulfate could also switch the CO₂ hydrogenation from methanation to reverse-water gas shift reaction. These results further suggested the ubiquitous role of residual sulfate in controlling the product selectivity in the CO₂ hydrogenation reaction.

We have added relevant description in this revised manuscript in lines 341-344 on page 19 “We also observed the similar product selectivity inversion by sulfate modification on other methanation catalysts such as Rh/TiO₂ and Ni/TiO₂ catalysts (Supplementary Figs. 16, 17). These results further suggested the ubiquitous role of residual sulfate in controlling the product selectivity in the CO₂ hydrogenation.”

Supplementary Fig. 16 | Temperature-dependent (a) CO₂ conversions and (b) CH₄ selectivity on Rh/TiO₂ catalysts containing or not containing sulfate species.

Supplementary Fig. 17 | Temperature-dependent (a) CO₂ conversions and (b) CH₄ selectivity on Ni/TiO₂ catalysts containing or not containing sulfate species.

4) The scale bar for DRIFT spectra in Figure 3c should be provided.

Reply: Thanks for your kind suggestions. We have added the scale bar in Figure 3c in this revised manuscript.

We sincerely hope this carefully revised manuscript can be accepted for publication now.

Reviewer #3:

In this manuscript, it was mainly discovered that air annealing at high temperature induced the migration of sulfates to the Ru-TiO₂ interface, resulting in the formation of RuS_x. The chemical bonding of Ru-S could facilitate electron transfer from Ru to S and enhance hydrogen spillover from Ru to TiO₂, thus weakening the adsorption and activation of CO intermediates and leading to low CO₂ conversion and high CO selectivity during CO₂ methanation. The manuscript is well-written, and the main conclusions that the authors highlight can be derived from the results. Before proceeding to publication, we recommend properly addressing the following points:

1. In Fig. 1a, CO was the main product on Ru/Ti-Aldrich-4 with 0.58% S content, contrasting with its high CH₄ selectivity in Supplementary Fig. 3b.

Reply: Thanks for your kind comments and reminder. We made some mistakes in the color and symbol for the activity curve of Ru/Ti-Aldrich-4 in Supplementary Fig. 3b. We have corrected these mistakes in this revised manuscript.

Supplementary Fig. 3 | (a) Temperature-dependent CO₂ conversions and (b) CH₄ selectivity on Ru/TiO₂ catalysts prepared using different commercial anatase TiO₂ supports.

2. According to thermodynamic equilibrium (Supplementary Fig. 2), CO₂ conversion can not reach 100% above 320 °C in Fig. 1b.

Reply: Thank you for your comments. The thermodynamic equilibrium CO₂ conversion vs. temperature has been shown in Supplementary Fig. 2, and CO₂ conversions at temperatures higher than 325 °C indeed could not reach 100%. By rechecking the raw data of the activity tests, we found that the standard curve of CO₂ concentration exhibited inadequate accuracy when CO₂ was at low concentrations. We rebuilt the standard curve of CO₂ concentrations and retested the activity of the samples, and the results showed that the CO₂ conversion on Ru/TiO₂-AR reached the highest of *ca.* 89% at 330 °C and then slightly decreased to 83% at 410 °C. All related data have been updated in this revised manuscript.

The relevant contents have been added into the revised manuscript in lines 116-119 on page 7 “Fig. 1b shows that the Ru/Ti-AR exhibited excellent activity for CO₂ hydrogenation between 150 and 410 °C, and the CO₂ conversion reached the highest of *ca.* 89% at 330 °C and slightly decreased to 83% at 410 °C. The CH₄ selectivity maintained above 95% within this temperature range.”

Fig. 1b. Temperature-dependent CO₂ conversion and CH₄ selectivity of Ru/TiO₂ catalysts with or without SO₄²⁻ species.

3. What temperature are the data in Fig. 1d and Supplementary Fig. 7 shown at? Neither figure caption nor main text specified it.

Reply: Thank you for your kind reminding. The temperature in Fig. 1d and the Supplementary Fig. 7 were both 350 °C, and we had added the temperature condition in the figure captions and main text.

4. In the case of sulfur-free Ru/rutile (Supplementary Fig. 5), air annealing prior to H₂ reduction did not have any influence on CO₂ conversion. How about in the case of sulfur-free Ru/anatase? Ru/Ti-AR and Ru/Ti-R do have different Ru particle size distributions, 5-12 nm and 1-7 nm, respectively. Did the difference in Ru particle sizes affect their CO₂ conversion and CO selectivity? A comparison of temperature-dependent CO₂ conversion/selectivity between

Ru/Ti-AR and Ru/Ti-R should be presented.

Reply: Thank you for your kind comments. Following your suggestion, we have presented the results about the comparison of temperature-dependent CO₂ conversion and product selectivity on the Ru/Ti-R (anatase) and Ru/Ti-AR (anatase) in Supplementary Fig. 8. The CO₂ conversion on Ru/Ti-R was higher than that on Ru/Ti-AR, while both the Ru/Ti-R and Ru/Ti-AR showed the high CH₄ selectivity. These results indicated that the difference in Ru particles on sulfur-free Ru/Ti-R (2.9 nm) and Ru/Ti-AR (5.5 nm) only had some impact on the CO₂ conversion, but did not influence the selectivity.

Supplementary Fig. 8 | (a) Temperature-dependent CO₂ conversions and (b) CH₄ selectivity on Ru/Ti catalyst with air and/or H₂ pretreatment.

5. Plots such as supplementary Fig. 4 (and some others) should be supplemented with data obtained at iso-conversion (or close to). It is too hasty to conclude on a selectivity inversion, when conversion levels are so different. For example, decreasing markedly the contact time for the CH₄-selective catalyst might result in much lower CO₂ conversion at high temperature and

possibly not so high CH₄ selectivity. Similarly, increasing contact time with the CO-selective catalyst will increase conversion; it would be interesting to check if CO selectivity remains so high in such case or if it moves towards CH₄. Authors should back up their strong claims with such control experiments.

Reply: Thank you for your kind comments and suggestion. We understand your concern that the selectivity inversion on Ru/Ti-S-AR might be due to its lower CO₂ conversion levels. Following your suggestion to check CO selectivity at high CO₂ conversion, we tested the activity of Ru/Ti-S-AR at different contact time by changing the weight hourly space velocity (WHSV). The results showed that CO₂ conversion was enhanced by increasing contact time, but high CO selectivity still remained, further confirming the high CO selectivity on Ru/Ti-S-AR catalyst.

Supplementary Fig 8. | Temperature-dependent CO₂ conversions and CO selectivity on Ru/Ti-S-AR catalysts when changing the weight hourly space velocity.

Some revisions have been added in the updated manuscript in lines 122-125 on page 7 “Besides, we tested the activity of the Ru/Ti-S-AR by altering contact time (Supplementary Fig. 5). The results showed that the CO₂ conversion was enhanced by increasing contact time, but high CO selectivity still remained, further confirming the high CO selectivity on Ru/Ti-S-AR.”

6. Fig. 1a is not very informative (also it contains symbols which are not explained in caption); it would be advisable to find a way to capture the essence of the Table reported in the supplementary, in a plot that show quantitative results. For example, CH₄ yield and CO yield as a function of S content?

Reply: Thank you for your kind comments. We accepted your advice that displaying the essence of the supplementary Table 1 (the sulfur content of different catalysts) in Fig. 1a, and some revisions have been made in the revised Fig.1a and its captions.

Fig. 1 | Catalytic performance of the Ru/TiO₂ catalysts. **a** The products and sulfur content comparison on the different sets of Ru/TiO₂ catalysts in CO₂ hydrogenation (TiO₂ support were purchased from Aldrich and Aladdin and were denoted as Ti-Aldrich and Ti-Aladdin, respectively). **b** Temperature-dependent CO₂ conversion and CH₄ selectivity of Ru/TiO₂ catalysts with or without SO₄²⁻ species. **c** Comparison with commercial Ru/Al₂O₃ catalyst for CH₄ productivity at 350 °C and commercial CuO/ZnO/Al₂O₃ catalyst for CO productivity at 410 °C. **d** The product selectivity on Ru/Ti-S catalyst with air and/or H₂ pretreatment at 350 °C.

7. It is a weakness of the submission that Ru particles size (= an important parameter!) is only assessed through electron microscopy. What is the number of particles counted for each size distribution reported? Ideally, dispersion should be discussed in quantitative terms, using chemisorption data, to prove beyond reasonable doubt that indeed Ru dispersion (related to nanoparticles size) is indeed similar with or without S contamination.

Reply: Thank you for your valuable comments. The results about the distribution of Ru particles were obtained by counting about 200 particles. Following your suggestion, we also tested the Ru dispersion by CO chemisorption and the results were shown in Supplementary Table 2. The presence of sulfate had little influence on the size distribution of Ru particles.

We understand that you are concerned about the influence of Ru particles size on the catalytic performance. Actually, previous studies had demonstrated that the high CH₄ selectivity was obtained on relatively large particles (> 1 nm) (*ACS Catal.* 2013, 3, 2449-2455; *ACS Catal.* 2018, 8, 6203-6215). Our Ru dispersion results clearly showed that the size of Ru particles with or without sulfate were similar to each other with a large average size of > 1 nm. In addition, our testing results also showed that the Ru/Ti-R, Ru/Ti-AR and Ru/Ti-R-AR with

different but > 1 nm Ru particles sizes presented high CH₄ selectivity (Supplementary Figs. 7, 8). Therefore, we can conclude that the selectivity inversion from high CH₄ selectivity to high CO selectivity were not related to the Ru particle size.

Some revisions have been added in the updated manuscript in lines 173-177 on page 10 “A summary about the Ru particles size and dispersion was shown in Supplementary Table 2. These results suggested that the air annealing at high temperature followed by H₂ reduction led to a higher degree of Ru particle aggregation compared with direct H₂ reduction treatment, and the presence of sulfate had negligible influence on the Ru particle size distribution.”

Supplementary Table 2 | Ru contents, Ru particle size and Ru dispersion on different Ru/TiO₂ catalysts.

	Ru contents ^a	Ru particle diameter ^b	Ru dispersion ^c
Ru/Ti-R	4.95%	2.9 nm	25.2%
Ru/Ti-AR	4.98%	5.5 nm	12.1%
Ru/Ti-S-R	4.94%	2.6 nm	26.7%
Ru/Ti-S-AR	4.95%	5.7 nm	10.6%

^a Determined from ICP results.

^b Calculated from TEM images.

^c Calculated from CO chemisorption results.

8. Rutile-based catalysts are briefly mentioned (line 130) and claimed to behave like anatase-based ones (according to catalytic activity data). It is known, however, that Ru on rutile TiO₂ can have a peculiar behavior during annealing in air, due to the epitaxial match between RuO₂ and rutile TiO₂. Did authors observe any kind of restructuration of the support (to be checked e.g. by TEM, N₂-physisorption, XRD) after annealing, and is this affected by the presence of S?

Reply: Thank you for your nice question. To exclude the influence of crystal structure of TiO₂ on the product selectivity, we have tested and compared the performance of the rutile-based catalysts with the anatase-based catalysts (Supplementary Figs. 6-8). It was shown that both the sulfate-free Ru/TiO₂ catalysts (using anatase or rutile) showed high CH₄ selectivity, while the sulfate-modified Ru/TiO₂ (using anatase or rutile) after annealing in air showed high CO selectivity. We can confirm by these results that the presence of sulfate residuals on TiO₂ supports, rather than the crystal structure of TiO₂, was the determining factor influencing the product selectivity.

You mentioned the possible restructuration of rutile TiO₂ due to the epitaxial match between RuO₂ and rutile TiO₂ during annealing in air. Since the high CH₄ selectivity was observed on both Ru/rutile-AR (with annealing in air) and Ru/rutile-R (without annealing in air), we believe that, even though there was some restructuration of rutile TiO₂ during annealing in air, it has a negligible impact on the product selectivity. Hence, we did not explore the restructuration of the rutile in this work but it may be done in future study. Hope you can understand us.

In addition, you are concerned that the presence of trace amount of sulfate might affect the

restructuring of the support. According to the XPS spectra of Ru/Ti-AR and Ru/Ti-S-AR (Supplementary Fig. 13), there were no obvious changes in Ti 2p, indicating that the presence of trace amount of sulfate might have negligible influence on the chemical states of surface Ti species and might not cause the restructuring of the TiO₂.

9. Line 219: The citation regarding the reduction of RuS_x is not accurate. Reference 30 investigated the effect of alkali additives (Li, Na, K, Cs) without the presence of sulfate species. The following references, which reported the reduction of amorphous RuS₂, may be helpful: <https://doi.org/10.1021/cs100053e> & <https://doi.org/10.1016/j.jcat.2008.08.015>

Reply: Thank you for your kind comments and advice. We made some mistake in the reference citation, and we have made careful revision according to your suggestions.

30. Infantes-Molina, A. *et al.* Role of Cs on hydrodesulfurization activity of RuS₂ catalysts supported on a mesoporous SBA-15 type material. *ACS Catal.* **1**, 175-186 (2011).

31. Castillo-Villalón, P. *et al.* Structure, stability and activity of RuS₂ supported on alumina. *J. Catal.* **260**, 65-74 (2008).

10. Line 231: a 0.2 eV shift in XPS is indeed a very slight shift. Please note that surface acidity (which can be affected by the presence of S) can cause different levels of C contamination, which itself can cause differential charging effects and issues with BE calibration (see <https://doi.org/10.1002/cphc.201300411> and <https://doi.org/10.1016/j.pmatsci.2019.100591>). It is advised to also try to apply a BE calibration on the Ti 2p peak, and to check if the 0.2 eV shift in the Ru 3d peak still holds.

Reply: Thank you for your kind suggestion. We have carried out BE calibration on the Ti 2p peak, and confirmed that the 0.2 eV shift in the Ru 3d peak still holds. We agree that this shift was related to the presence of S, which induced the interaction between Ru particles and sulfate at the Ru-Ti interface. Since the amount of S content was very low (0.2%), only slight shift in the Ru 3d peak was observed.

Supplementary Fig. 12 | Ru 3d XPS of Ru/Ti-AR and Ru/Ti-S-AR.

11. Line 234: The authors stated that the presence of sulfate showed no influence on the chemical states of surface Ti species. It would be better to elucidate that the presence of sulfate at Ru-TiO₂ interface could enhance hydrogen spillover from Ru to TiO₂, thereby creating Ti³⁺ shallow traps, as later revealed by in-situ DRIFTS under H₂. However, XPS may not be sensitive enough to observe this change in chemical states of Ti species.

Reply: Thank you for your kind comments. We have investigated the chemical states of Ru and Ti by XPS (Supplementary Figs. 12 and 13) to explore the state changes of Ru and Ti species with and without sulfate on Ru-TiO₂. There were no obvious changes in Ti 2p XPS of Ru/Ti-AR and Ru/Ti-S-AR (Supplementary Fig. 13), indicating that the presence of low amount of S (0.2%) had no clear influence on the chemical states of surface Ti species. We agree with you that the presence of sulfate at Ru-TiO₂ interface could enhance H spillover from Ru to TiO₂, thereby creating Ti³⁺ shallow traps, which was revealed by in-situ DRIFTS, while XPS may not be sensitive enough to observe this change in chemical states of Ti species under H₂.

We have made some revisions in this updated manuscript in lines 232-237 on page 13-14 “Specifically, the presence of the sulfate induced a slight shift of the metallic Ru peak from 279.7 to 279.9 eV, which might be due to the formation of Ru-S bonds at the Ru-Ti interface. For the impact of sulfate species on the structure TiO₂, no obvious changes were observed in Ti 2p on Ru/Ti-AR and Ru/Ti-S-AR (Supplementary Fig. 13), indicating that the presence of trace amount of sulfate species had negligible influence on the TiO₂ support.”

12. Line 241: Reference 20 only assigned the bands at 2128 and 2068 cm⁻¹ to vibrations of multicarbonyl species on undercoordinated Ru atoms and CO on-top of Ru atoms, respectively. The band at 2178 cm⁻¹ was not present in the CO-DRIFTS spectra of Ru/ γ -Al₂O₃ in reference 20. Could it be possible that the band at 2178 cm⁻¹ was attributed to CO adsorption on Ti⁴⁺ sites (<https://doi.org/10.1038/s41467-022-32934-5>)? Reasonably, this band disappeared in the CO-DRIFTS spectra of sulfate-containing Ru/TiO₂ catalysts (Fig. 3c).

Reply: Thank you for your kind suggestions. We carefully checked the literature (*Nat. Com.* 2022, 13, 5186; *Catal. Com.* 2007, 8, 1715-1718), and we agree with you that the band at 2178 cm⁻¹ was attributed to CO adsorption on Ti⁴⁺ sites. We have added some discussions in the revised manuscript in lines 246-247 on page 14 “In addition, the band at 2178 cm⁻¹ was attributed to CO adsorption on Ti⁴⁺ sites³⁴.” and lines 252-254 on page 14 “With the introduction of the sulfate on TiO₂, the CO adsorbed on TiO₂ (2178 cm⁻¹) nearly disappeared on Ru/Ti-S-R and Ru/Ti-S-AR, which should be due to the covering of these surface cationic Ti sites by sulfate.”

13. Line 331 mentions the case of Al₂O₃ supported Ru catalysts and claims similar trends were observed (as compared to the TiO₂-supported ones). In the case of Al₂O₃, one should rather talk about a complete deactivation of the catalyst, in the presence of S! In this case also, it would be advisable to show the data for the full temperature range, for Ru/Al (CH₄ selectivity appears to start to drop at 375°C...)

Reply: Thank you for your comments. We carefully reanalyzed the activity results of Ru/Al-S and Ru/Si-S, and noted that significant deactivation of these two catalysts before 500 °C, which was indeed not a similar trend compared with Ru/Ti-S. Thus, our claims that the universal role of residual sulfate on supports in controlling the product selectivity in the CO₂ hydrogenation might be not reasonable, hence we deleted these results in the revised supplementary materials.

According to the suggestions of Reviewer #2, the universality to tune the selectivity for CO₂ hydrogenation by sulfate species could be verified using other typical catalyst for methanation (Rh and Ni). We prepared the Rh/Ti-AR, Rh/Ti-S-AR, Ni/Ti-AR and Ni/Ti-S-AR, and tested their activities. The results are presented in Supplementary Fig. 17 and 18. It was found that trace amounts of sulfate on Rh/TiO₂ and Ni/TiO₂ catalysts could also totally change the CO₂ hydrogenation from methanation to reverse-water gas shift reaction, indicating the similar trend compared with Ru/TiO₂ catalyst. These findings further suggested the ubiquitous role of residual sulfate in controlling the product selectivity in the CO₂ hydrogenation.

We understand that you are concerned with the temperature-dependent of CO₂ conversions and CH₄ selectivity at full temperature range on Ru/Al catalysts. We present the data here for your convenience in Fig. R1. Actually, both CO₂ conversion and CH₄ selectivity dramatically dropped above 400 °C. The deactivation or decreased activity on Ru/Al might be due to many factors, and we will conduct a detailed discussion in our following work. Hope you can understand us.

We have added relevant description in the revised manuscript in lines 341-344 on page 19 “We also observed the similar product selectivity inversion by sulfate modification on other methanation catalysts such as Rh/TiO₂ and Ni/TiO₂ catalysts (Supplementary Figs. 16, 17). These results further suggested the ubiquitous role of residual sulfate in controlling the product selectivity in the CO₂ hydrogenation.”

Again, thank you for your valuable comments and suggestions on our manuscript, which helped us improve the quality significantly. We sincerely hope that this carefully revised manuscript can be accepted for publication now.

Supplementary Fig. 16 | Temperature-dependent (a) CO₂ conversions and (b) CH₄ selectivity on Rh/TiO₂ catalysts containing or not containing sulfate species.

Supplementary Fig. 17 | Temperature-dependent (a) CO₂ conversions and (b) CH₄ selectivity on Ni/TiO₂ catalysts containing or not containing sulfate species.

Supplementary Fig. R1 | Temperature-dependent (a) CO₂ conversions and (b) CH₄ selectivity on Ru/Al catalysts containing or not containing sulfate species.

Point-to-Point Responses to the Reviewers' Comments

Reviewer #1

In the revised manuscript the authors have corrected their data to support their arguments of promoted H-spillover by sulfate to change CO₂ hydrogenation product from CH₄ to CO. This research basically reports a phenomenon that sulfate modification can alter the product selectivity, but the discussion about underlying structure-function relationship is not correct at all. Overall, their argument cannot be effectively supported by the data. Many claims made in the manuscript are actually assumptions and guesses rather than conclusions derived from solid data supports. I do not support further consideration for publication on Nature communications. Some critical comments are listed here.

1. The TiO₂ supports used in this work should be anatase, although no XRD and Raman spectra have been provided in the main text and supporting information. It is well acknowledged by catalysis community that anatase TiO₂ is the most typical oxide to display hydrogen spillover effect. I do believe sulfate can promote H-spillover enough to change the product reactivity. There should be other changes in the reaction kinetics. Although the authors have conducted DRIFTS and DFT calculations to support promoted H-spillover, these methods actually cannot provide solid supports for H-spillover. They should clarify the mechanisms by studying how CO₂ is activated and converted by the catalysts. Unfortunately, no characterization and discussion on the reaction mechanisms, so it is rude to ascribe the mechanisms to H-spillover without sufficient experimental supports.

Reply: Thank you for your detailed comments. The TiO₂ used in our work was synthesized by hydrolyzing tetrabutyl titanate, followed by the calcination in muffle furnace at 400 °C for 2 h. This was a typical method to obtain anatase TiO₂. According to your suggestion, we also determined the crystal phases of the prepared Ru/Ti catalysts by XRD (please see Supplementary Fig. 12a). The most prominent diffraction peaks were detected at 25.3 and 48.1 °, which was related to the (101) and (200) planes of anatase. Raman spectra results (please see Supplementary Fig. 13) show that all Ru/Ti catalysts exhibited vibrational modes at 143, 395, 515 and 638 cm⁻¹, which were in accordance with the characteristics of anatase phase.

Your main concern on our work might be the mechanism involving CO₂ conversion to CO and CH₄ over Ru/Ti-S-AR and Ru/Ti-AR catalysts. Several recent studies on Ru-based catalysts concurred that CO₂ was firstly converted to CO, followed by further hydrogenation of the chemisorbed CO to form CH₄ (*J. Am. Chem. Soc.* 2021, 143, 11582-11594; *Angew. Chem. Int. Ed.* 2020, 59, 19983-19989; *J. Am. Chem. Soc.* 2018, 140, 13736-13745). For instance, as shown in Supplementary Scheme R1, the C=O bonds in CO₂ were cleaved via direct interactions with exposed Ru atoms, which was facile in elementary steps. Such CO₂ activation steps resulted in the formation of bound CO molecules and O atoms; the latter were removed via H-addition steps to form H₂O. The kinetic hurdles in forming CH₄ required the cleavage of strong C≡O bonds in chemisorbed CO via hydrogen-assisted activation pathways (*J. Am. Chem. Soc.* 2021, 143, 11582-11594).

Following your suggestion, we further clarified the mechanisms involved in CO₂ activation and hydrogenation process over the catalysts by *in situ* DRIFTS experiments under steady-state CO₂ hydrogenation conditions. Two observations could be drawn from *in situ* DRIFTS results

(please see Supplementary Fig. 18): (1) CO₂ was easily converted to CO on Ru/Ti-AR and Ru/Ti-S-AR, even at very low temperature (323 K). (2) The intermediate CO adsorbed at Ru site of Ru/Ti-AR could be converted to CH₄ when the reaction temperature was above 473 K, while the intermediate CO adsorbed at Ru site of Ru/Ti-S-AR was stable and no CH₄ was obtained. In combination with H₂-DRIFTS and DFT calculation results suggesting that the surface sulfate species at the interfacial Ru sites greatly enhanced the H₂ spillover process, we claimed that the sulfate modification on Ru catalysts promoted the H transfer from Ru particles to TiO₂, inducing the switch from high CH₄ selectivity to high CO selectivity. We hope that our new experimental results and explanations adequately address your concerns regarding the reaction mechanism.

Accordingly, we made some revisions in the updated manuscript in lines 326-331 on page 19: “We performed DRIFTS studies under steady-state CO₂ hydrogenation conditions, and it was observed that CO₂ was easily converted to intermediate CO on Ru/Ti-AR and Ru/Ti-S-AR at low temperature. Additionally, the intermediate CO adsorbed at Ru site of Ru/Ti-AR could be converted to CH₄ when the reaction temperature was above 473 K, while the intermediate CO adsorbed at Ru site of Ru/Ti-S-AR was stable and no CH₄ was obtained (Supplementary Fig. 18).” We also added some notes into the Supporting information on pages S19-S20 “*In situ* DRIFT spectra under steady-state CO₂ hydrogenation conditions are shown in **Supplementary Fig. 18**. On both Ru/Ti-AR and Ru/Ti-S-AR, CO stretching vibration bands on Ru (2010-2015 cm⁻¹) were observed immediately upon exposure to the feed gas, associated with the appearance of bicarbonates (HCO₃⁻, 1630 cm⁻¹), formates (HCO₂⁻, 1551 and 1354 cm⁻¹) and carbonates (CO₃²⁻, 1424 cm⁻¹) species. The bicarbonates were less stable and disappeared completely at 473 K. The intensity of the bands of formates and carbonates exhibited a limited change above 373 K. Additionally, it was observed that the intermediate CO adsorbed at Ru site of Ru/Ti-AR could be converted to CH₄ (3018 cm⁻¹) when the reaction temperature was above 473 K, while the intermediate CO adsorbed at Ru site of Ru/Ti-S-AR was stable and no CH₄ was obtained.”

Supplementary Fig. 12 | (a) Powder X-ray diffraction patterns of TiO₂, Ru/Ti-R, Ru/Ti-AR, Ru/Ti-S-R and Ru/Ti-S-AR.

Supplementary Fig. 13 | The Raman spectra of TiO₂, Ru/Ti-R, Ru/Ti-AR, Ru/Ti-S-R and Ru/Ti-S-AR.

Supplementary Scheme R1 | Elementary steps for CO₂-H₂ reactions that form CO and CH₄ on Ru surfaces.

Supplementary Fig. 18 | *In situ* DRIFT spectra following exposure to CO₂ and H₂ at 323 K and subsequent stepwise heating to 623 K for (a) Ru/Ti-AR and (b) Ru/Ti-S-AR.

2. Some claims in the this paper are not correct or misleading. “The enhanced hydrogen spillover weakened the activation of CO intermediates, leading to significantly higher selectivity for CO production.” H-spillover does not definitely change selectivity from CH₄ to CO, but sometimes can also enhance methanation by changing reaction pathways.

Reply: Thank you for your comments. We agreed with you that H spillover does not definitely change the selectivity from CH₄ to CO, but sometimes can also enhance methanation by changing reaction pathways. Indeed, H spillover represents the H transfer process from metal to support, which can prevent the deep hydrogenation of CO₂ on metal sites due to the lack of H atoms, or can promote the deep hydrogenation of CO₂ on the support owing to the accumulation of H atoms. In our present work, on Ru/TiO₂ catalysts, the CO₂ conversion to CO mainly occurred on Ru sites at low temperature (please see Supplementary Fig. 18). Hence, the possibility of H spillover inducing the enhancement of methanation on support could be excluded. On the Ru/Ti-AR, with the low extent of H spillover, the intermediate CO adsorbed at Ru sites was readily converted to CH₄ with the assistance of sufficient H on Ru sites. By

contrast, on the Ru/Ti-S-AR, due to the high extent of H spillover, the methanation of intermediate CO adsorbed was greatly prohibited, resulting in the selectivity change from CH₄ to CO.

Supplementary Fig. 18 | *In situ* DRIFT spectra following exposure to CO₂ and H₂ at 323 K and subsequent stepwise heating to 623 K for (a) Ru/Ti-AR and (b) Ru/Ti-S-AR.

3. It is very puzzling to form RuS_x through annealing sulfate with Ru/TiO₂ in air. What is the reducing reagent to reduce sulfate? The species of RuS_x actually are not supported by any solid data.

Reply: Thank you for your comments and question. Actually, we have mentioned in the manuscript that, by annealing Ru/TiO₂ containing sulfate in air first followed by H₂ reduction, the RuS_x was formed at the interface of Ru/TiO₂. The reducing reagent was H₂. We demonstrated the presence of RuS_x by EDX mapping (determining the distribution of sulfur species on the surface of Ru/Ti-S-AR and Ru/Ti-S-R) and XPS (determining chemical states of sulfur species). From the EDX mapping results, sulfur element tended to accumulate around Ru species on Ru/Ti-S-AR (please see Fig. 2b), while the sulfur element was only randomly distributed on Ru/Ti-S-R (please see Fig. 2d). This suggested that the sulfur species on TiO₂ tended to migrate to Ru particles under air-H₂ treatment (Ru/Ti-S-AR), while it randomly

migrated on TiO₂ under direct H₂ treatment (Ru/Ti-S-R). From the XPS results (please see Fig. 3a), the appearance of the peaks at 161.5 and 162.5 eV confirmed the presence of S²⁻ on Ru/Ti-S-AR, implying the formation of Ru-S bonds, and the intensity of these two peaks were considerably stronger compared to that on Ru/Ti-S-R. We believed that the results of EDX mapping and XPS could well support the presence of RuS_x.

Fig. 2 | The geometric states of Ru NPs on TiO₂. **a** HAADF-STEM image of Ru/Ti-S-AR. **b** EDX mapping images of Ru and S elements on Ru/Ti-S-AR. **c** HAADF-STEM image of Ru-Ti-S-R. **d** EDX mapping images of Ru and S elements on Ru/Ti-S-R. **e** Schematic illustration of the evolution of Ru and S species on TiO₂ during air-H₂ or direct H₂ treatment.

Fig. 3a S 2p XPS for Ru/Ti-S-AR, Ru/Ti-S-R, Ru/Ti-AR and Ru/Ti-R.

4. “We conducted a comprehensive in situ DRIFTS study of H₂ reactions with different samples to investigate the migration of H atoms and electrons.” This method is not right. In situ DRIFTS probe the activation and intermediates of CO₂ but not H migration. It is very unusual that the FTIR peaks are so broad. The measurement conditions should be not well optimized before collecting spectra.

Reply: Thank you for your comments. In the process of H spillover, H₂ molecules dissociated into H atoms at metallic sites and spilled over to O sites on the surface of TiO₂ forming localized Ti-O(H)-Ti species. Simultaneously, the electrons were donated into the shallow trap states in the band gap of TiO₂, leading to a broad IR absorbance in the spectrum. The broad IR absorbance induced by the accumulation of electrons had been reported in the literatures (*J. Phys. Chem. C* 2012, 116, 4535-4544). An example was provided in Supplementary Fig. R1, and the IR spectrum response the shallow-trapped electrons on TiO₂ was observed (*J. Phys. Chem. C* 2012, 116, 4535-4544). This method has been reported to determine the extent of H and electrons migration (*Angew. Chem. Int. Ed.* 2015, 54, 5905-5909). Hence, it was reasonable that we used this method to investigate the migration of H atoms and electrons. Hope this well addresses your concern.

Supplementary Fig. R1 | The IR spectrum response shallow-trapped electrons on TiO₂.

Reviewer #2

The authors have addressed all my comments satisfactorily. Especially, the new supplementary experiments on other methanation metal catalysts such as Rh and Ni have verified its universality to tune the selectivity for CO₂ hydrogenation by sulfate species, providing broad implications for the design and development of more efficient and selective heterogeneous catalysts. Thus, I recommend its publication in Nature Communications.

Reply: We highly appreciate your positive comments on our work. Thank you very much.

Reviewer #3

We stand by our comments that the paper is innovative, interesting, and potentially impactful. We acknowledge the efforts made by the authors to respond to reviewers comments, and in particular to solve some of the issues we had pointed earlier.

Based on their response, however, we still have some reservations on some of our comments, and we add two new comments:

Comment 2: The authors stated in their response that the activity of the samples was “retested”. However, the CO₂ conversion data at low temperatures appear exactly identical to those in the original manuscript; only the data at higher temperatures (> 330 °C) are lower, following the thermodynamic equilibrium CO₂ conversion. Are the data presented in this figure obtained from different experiments? (previous experiment for low T, and new experiment for high temperature?). Error bars could be added.

Reply: Thank you for your kind comments. In our previous experiment, the standard curve of CO₂ concentration exhibited inadequate accuracy when CO₂ was at low concentrations. We rebuilt the standard curve of CO₂ concentrations and retested the activity of the samples, and the data in all figures are from new experiments. Indeed, the obtained data from new experiment at low temperatures are almost identical to the previous data, but the data at high temperatures are slightly lower. We calculated the experimental errors for the rate and selectivity measurements, and the errors were consistently less than $\pm 5\%$. We tried to add error bars, however, due to the narrow spacing between the lines, the error bars on the lines seriously overlapped. Hence, we did not add the error bars in the data figures, but we added the information on the error bars to the experimental section of our revised manuscript in line 420-421 on page 23: “The experimental data variability for the activity tests was less than $\pm 5\%$ ”.

Comment 5: Increasing the contact time can be achieved by reducing the flow rate of the feed or increasing the amount of catalyst. Both result in a decrease in WHSV. In Supplementary Fig. 8, increasing the contact time (lower WHSV) leads to lower CO₂ conversion; this is not logical and may be a mistake. The authors should check the accuracy of the data.

Reply: Thank you for your kind comments and reminder. We made some mistakes in the color and symbol in Supplementary Fig. 5. We have corrected these mistakes in this revised manuscript.

Supplementary Fig. 5 | Temperature-dependent CO₂ conversions and CO selectivity on

Ru/Ti-S-AR catalysts when changing the weight hourly space velocity.

Comment 6: Fig. 1a still contains the names of the different catalysts in front of colored symbols; but these are not used in the rest of the figure. Why showing a list of samples, if one does not discuss the differences between them. Only the dichotomy between high and low S content is useful here.

Reply: Thank you for your kind suggestions. We deleted the list of samples and directly displayed the products and sulfur content comparison in Fig. 1a.

Fig. 1 | Catalytic performance of the Ru/TiO₂ catalysts. a The products and sulfur content comparison on the different sets of Ru/TiO₂ catalysts for CO₂ hydrogenation. **b** Temperature-dependent CO₂ conversion and CH₄ selectivity of Ru/TiO₂ catalysts with or without SO₄²⁻ species. **c** Comparison with commercial Ru/Al₂O₃ catalyst for CH₄ productivity at 350 °C and commercial CuO/ZnO/Al₂O₃ catalyst for CO productivity at 410 °C. **d** The product selectivity on Ru/Ti-S catalyst with air and/or H₂ pretreatment at 350 °C.

Comment 7: The characterization section should also include details on CO chemisorption measurement.

Reply: Thank you for your suggestion. The CO pulse chemisorption dispersion measurements were performed using a Micromeritics AutoChem II 2920 gas analyzer. In a typical measurement, approximately 70 mg of calcined catalyst was loaded into a U-shaped sample tube and reduced at 573 K for 1 h in 10% H₂/Ar. The catalyst was then flushed with He for 30 min. After cooling the sample to 323 K, pulse chemisorption measurements were performed

with 10% CO/He while monitoring the effluent with a thermal conductivity detector.

We have added relevant description in this revised manuscript in lines 403-407 on page 23 “For CO pulse chemisorption dispersion measurements, 70 mg of calcined catalyst was loaded into a U-shaped sample tube and reduced at 673 K for 1 h in 10% H₂/Ar. The catalyst was then flushed with He for 30 min. After cooling the sample to 323 K, pulse chemisorption measurements were performed with 10% CO/He while monitoring the effluent with a TCD.”

Comment 8: we were expecting an answer based on XRD results (to check re-structuration of the crystalline phases), not XPS.

Reply: Thank you for your comments. Following your kind suggestion, we performed XRD measurements to determine the crystal phases of Ru/Ti and Ru/rutile catalysts. As shown in Supplementary Fig. 12, for Ru/Ti samples, the most prominent diffraction peaks were detected at 25.3 and 48.1 °, which was related to the (101) and (200) planes of anatase. For Ru/rutile samples, 27.5 and 36.2 ° were the most prominent diffraction peaks, which were ascribed to the (110) and (101) planes of rutile. These results demonstrated that sulfate species did not induce the restructuration of the support.

Some revisions have been added in the updated manuscript in lines 237-239 on page 14 “the presence of trace amount of sulfate species had negligible influence on the TiO₂ support, which was in line with the XRD and Raman results (Supplementary Figs. 12,13).” We also added some notes in Supporting information on page S14, “The powder X-ray diffraction patterns of the prepared Ru/Ti and Ru/rutile samples are shown in Supplementary Fig. 12. The most prominent diffraction peaks were detected at 25.3 and 48.1 °, which was related to the (101) and (200) planes of anatase. For all Ru/rutile samples, 27.5 and 36.2 ° were the most prominent diffraction peaks, which were ascribed to the (110) and (101) planes of rutile. Hence, sulfate species did not induce the restructuration of the support. In addition, the diffraction peak at $2\theta = 44.0^\circ$ on Ru/Ti-R, Ru/Ti-AR, Ru/Ti-S-R and Ru/Ti-S-AR could be due to the diffraction of the (101) planes of Ru. We calculated the average Ru crystallite size using the Scherrer equation and the results were 10.5, 13.1, 11.3, and 15.4 nm, respectively. It should be pointed out that the Ru peaks were overlapped by the (210) plane of rutile TiO₂. Hence, we did not calculate the average crystallite size of Ru species on Ru/rutile catalysts by XRD.”

Supplementary Fig. 12 | Powder X-ray diffraction patterns of **(a)** TiO₂, Ru/Ti-R, Ru/Ti-AR, Ru/Ti-S-R and Ru/Ti-S-AR; **(b)** Rutile, Ru/rutile-R, Ru/rutile-AR, Ru/rutile-S-R, Ru/rutile-S-AR.

New comments:

1. The characterization section provides details on XRD analyses; however, no XRD results are presented and discussed. Showing XRD patterns could be useful, providing direct or indirect evidence (e.g., crystalline phases of anatase or rutile TiO₂, any possible phase transformation/restructuring, and size of Ru species).

Reply: Thank you for your kind comments. In this updated manuscript, we have included the XRD results, and the detailed explanation and responses to your questions can be found in the reply to your previous Comment 8. Accordingly, some revisions have been added in this updated manuscript in lines 172-173 on page 10: “The sizes of Ru derived from HAADF-STEM were in good agreement with XRD data (Supplementary Fig. 12).” lines 237-239 on page 14: “the presence of trace amount of sulfate species had negligible influence on the TiO₂ support, which was in line with the XRD and Raman results (Supplementary Figs. 12,13).” We also added some notes in supporting information on page S14: “The powder X-ray diffraction patterns of the prepared Ru/Ti and Ru/rutile samples are shown in Supplementary Fig. 12. The

most prominent diffraction peaks were detected at 25.3 and 48.1 °, which was related to the (101) and (200) planes of anatase. For all Ru/rutile samples, 27.5 and 36.2 ° were the most prominent diffraction peaks, which were ascribed to the (110) and (101) planes of rutile. Hence, sulfate species did not induce the restructuration of the support. In addition, the diffraction peak at $2\theta = 44.0^\circ$ on Ru/Ti-R, Ru/Ti-AR, Ru/Ti-S-R and Ru/Ti-S-AR could be due to the diffraction of the (101) planes of Ru. We calculated the average Ru crystallite size using the Scherrer equation and the results were 10.5, 13.1, 11.3, and 15.4 nm, respectively. It should be pointed out that the Ru peaks were overlapped by the (210) plane of rutile TiO₂. Hence, we did not calculate the average crystallite size of Ru species on Ru/rutile catalysts by XRD.”

Supplementary Fig. 12 | Powder X-ray diffraction patterns of (a) TiO₂, Ru/Ti-R, Ru/Ti-AR, Ru/Ti-S-R and Ru/Ti-S-AR; (b) Rutile, Ru/rutile-R, Ru/rutile-AR, Ru/rutile-S-R, Ru/rutile-S-AR.

Supplementary Table 2 | Ru contents, Ru particle size and Ru dispersion on different Ru/TiO₂ catalysts.

	Ru contents ^a	Ru particle diameter ^b	Ru crystallite size ^c	Ru dispersion ^d
Ru/Ti-R	4.95%	2.9 nm	10.5 nm	25.2%
Ru/Ti-AR	4.98%	5.5 nm	13.1 nm	12.1%
Ru/Ti-S-R	4.94%	2.6 nm	11.3 nm	26.7%
Ru/Ti-S-AR	4.95%	5.7 nm	15.4 nm	10.6%

^a Determined from ICP results.

- ^b Calculated from HAADF-STEM images.
- ^c Calculated from XRD.
- ^d Calculated from CO chemisorption results.

2. Was the H₂-TPR experiment performed without pre-reduction under H₂? The catalysts' name in H₂-TPR profiles might be misleading. Should they be changed to Ru/Ti, Ru/Ti-A, Ru/Ti-S and Ru/Ti-S-A?

Reply: Thank you for your kind suggestion. The H₂-TPR experiment was performed without pretreatment of H₂ reduction. Based on your advice, we changed the names of the catalysts in H₂-TPR profiles, as shown below.

Fig. 3 | b H₂-TPR profiles of Ru/Ti, Ru/Ti-A, Ru/Ti-S and Ru/Ti-S-A.

Reviewer #4

This revised paper can be accepted, because moderate modifications were made.

Reply: We greatly appreciate your positive support on our work.

Point-to-Point Responses to the Reviewers' Comments

Reviewer #1

The authors have provided more data in the revised manuscript and addressed reviewers' issues. Although other reviewers have thought this paper can be accepted for publication, I still strongly insist to my initial suggestion that this paper is not acceptable for publication for the academic credits of the authors and Nature Communications. They are many questionable data and misleading claims in this submitted manuscript.

1. In the first submitted version CO₂ conversions are 100 % at temperature higher than 300 °C. This is not owing to the standard curve at all, because standard curves can only affect the values of CO₂ conversion and CH₂/CO selectivity. This makes me conclude some catalytic results are fabricated but not really measured. The corresponding authors should double check the raw data.

Reply: Thank you for your comments. Your concern is on the validity of all our activity testing results. Each sample was tested at least three times to verify the observed trends in our experiments. We have carefully checked the raw data again about the activity testing results. Table R1 and Table R2 show the raw data on the catalytic performance of Ru/Ti-AR and Ru/Ti-S-AR for your reference. The main product was CH₄ on Ru/Ti-AR, while CO was predominant on Ru/Ti-S-AR. All catalytic results were measured, not fabricated.

It appears that some inaccuracies of the activity results due to the initial standard curve issue in the first submitted version might have led to an unfavorable judgment on your side. However, after three rounds of careful revisions, we believe that this manuscript has been significantly improved.

Table R1. The raw data for the catalytic performance of Ru/Ti-AR.

Temperature (°C)	CO ₂ Concentration (ppm)	CH ₄ Concentration (ppm)	CO Concentration (ppm)
210	36620	3335	45
230	33239	6702	58
250	28732	11199	69
270	21972	17828	201
290	12958	26705	337
310	6107	33439	454
330	4400	35293	307
350	4800	34985	215
370	5200	34594	206
390	6000	33461	539
410	6800	31927	1273

Table R2. The raw data for the catalytic performance of Ru/Ti-S-AR.

Temperature (°C)	CO ₂ Concentration	CH ₄ Concentration	CO Concentration
----------------------------------	----------------------------------	---------------------

	(ppm)	(ppm)	(ppm)
210	39437	12	543
230	39437	10	550
250	38873	99	1100
270	38873	110	1127
290	37746	114	2254
310	37183	119	2817
330	36056	125	3845
350	34930	134	4949
370	33239	151	6609
390	31549	168	8282
410	29859	197	9944

2. The authors claim S on the surface promotes H transfer and selectivity change. In the experimental section, no descriptions about how TiO₂ supports were prepared from tetrabutyl titanate and how the catalysts were modified with S were given. It is very ridiculous the most important experimental details are not described at all.

Reply: Thank you for your comments. The preparation of TiO₂ through the hydrolysis of tetrabutyl titanate (TBOT) is a well-established method in literature; therefore, only brief details were provided in the Methods section in our initial manuscript. Regarding the information on sulfur addition, a description was already included in the lines of 109-113 on Page 6: “To further investigate the influence of SO₄²⁻ on the catalytic performance of CO₂ hydrogenation, we prepared the sulfate-free Ru/TiO₂ catalysts, in which the sulfate-free TiO₂ were synthesized by hydrolyzing tetrabutyl titanate, and also prepared the Ru/TiO₂ catalysts containing sulfate by purposely adding ammonium sulfate during the preparation process (with mole ratio of S/Ru set as 0, 0.03, 0.05, and 0.1)” (now marked blue in the main text).

Following your suggestions, we have now included more comprehensive details on both the preparation of TiO₂ and the addition of sulfate in the Methods section. Relevant descriptions have been added to the revised manuscript in lines of 377-386 on Page 22: “TiO₂ were synthesized by hydrolyzing TBOT in a mixture of anhydrous ethanol and distilled water with a molar ratio of n(TBOT)/n(C₂H₅OH)/n(H₂O) =1:15:4. Distilled water was dropped into the mixture of TBOT and anhydrous ethanol. The obtained precipitates were next dried and calcinated at 400 °C for 2 h. All Ru/TiO₂ catalysts with 5 wt.% Ru were prepared using the impregnation method. TiO₂ supports and a certain amount of Ru(NO₃)₃ were mixed in distilled water with stirring. The solution was evaporated at 60 °C under vacuum until dry. The resulting samples were dried at 100 °C overnight and then were calcined at 400 °C for 2 h in air or directly reduced with H₂ to prepare various Ru/TiO₂ catalysts. For the samples with sulfate addition, ammonium sulfate was incorporated during the impregnation process (with mole ratio of S/Ru set as 0, 0.03, 0.05, and 0.1).”

3. The authors ascribe the selectivity change to H spillover promoted by S. This is just a guess. Anatase TiO₂ is the most typical oxide support to display H spillover and strong metal-support interaction (SMSI). For Ru/TiO₂-anatase, even without S, H spillover is still an indispensable

process in CO₂ hydrogenation reaction. Instead, many researches have shown SMSI can stably occur for Ru/TiO₂-anatase in CO₂ hydrogenation reaction, and can change the product from CH₄ to CO. This phenomenon is not found in the data shown by this paper. This further leads me to doubt the data. I cannot agree with explanation about the selectivity change by H transfer. This will mislead future researches in this area.

Reply: Thank you for your detailed comments. Your first concern is regarding the attribution of the selectivity changes to the enhancement of H transfer caused by the sulfate modification. Several studies have demonstrated that H transfer significantly influenced the selectivity toward CH₄ or CO (*Angew. Chem. Int. Ed.* 2020, 59, 19983; *Nature. Commun.* 2022, 13, 327). For instance, Qiao *et al.* revealed that a weak hydrogen spillover from Ru to TiO₂ favored the formation of CH₄, whereas an extensive hydrogen spillover shifted the selectivity change from CH₄ to CO (*Angew. Chem. Int. Ed.* 2020, 59, 19983). Notably, in the second review round of our manuscript, you also mentioned in your Comments #1: “I do believe sulfate can promote H-spillover enough to change the product reactivity.” In our present studies, we focused on how the sulfate modification changed the product from CH₄ to CO. The H₂-DRIFTS experiments and DFT calculations confirmed that the sulfate modification greatly enhanced the H transfer process at the Ru-TiO₂ interface. This is not questionable at all. Furthermore, following your suggestion in the second round of review, the mechanisms involved in the CO₂ hydrogenation process on Ru/Ti-AR and Ru/Ti-S-AR were studied by *in situ* DRIFTS. We observed that the intermediate CO adsorbed at Ru site of Ru/Ti-AR could be converted to CH₄ when the reaction temperature was above 473 K, whereas the intermediate CO adsorbed at Ru site of Ru/Ti-S-AR remained stable with no CH₄ formation. Based on these findings, we confidently proposed that the sulfate modification on Ru catalysts promoted H transfer, inducing a switch from high CH₄ selectivity to high CO selectivity. We hope this explanation resolves the concern that you may still have at this point.

Your second concern is about the impact of strong metal-support interaction (SMSI) in Ru/TiO₂-anatase on the product selectivity in CO₂ hydrogenation reaction. We agree with you that the SMSI in Ru/TiO₂ has the potential to change product from CH₄ to CO. Recent research found that annealing Ru/anatase-TiO₂ induced the SMSI and converted the product from CH₄ to CO (*Nature. Commun.* 2022, 13, 327). However, in our current studies, the primary focus is on the effect of sulfate modification on Ru/anatase-TiO₂ and its influence on the change in selectivity. This has no contradiction with the reported literature. Moreover, XRD, HAADF-STEM imaging, and XPS results show that the addition of trace amounts of sulfate species did not induce significant SMSI. Consequently, SMSI was unlikely to be a key factor influencing the product selectivity in our research. We hope this concern has now been sufficiently addressed to your satisfaction.

Reviewer #3

Authors have provided satisfactory responses to my questions and comments. The corresponding changes in the MS solve the issues I was pointing. I have also checked the additional data provided in response to other reviewer's comments, and found them both useful and valuable. I recommend this paper for publication.

Reply: We greatly appreciate your positive support on our work.

Point-to-Point Responses to the Reviewers' Comments

Reviewer #1

The authors have made commendable efforts to substantiate their claims and results, although the conclusions are not strongly supported by current data. This can be a very excellent study that presents significant findings in the field of heterogeneous catalysis for CO₂ hydrogenation if the mechanisms can be clearly revealed. To convincingly demonstrate that sulfur (S) influences selectivity by promoting hydrogen spillover rather than through a strong metal-support interaction (SMSI) effect, I recommend that the authors provide operando FTIR data in the main text rather than in the supporting information. Additionally, a detailed discussion on how sulfur alters the catalytic pathways would strengthen the manuscript (see *Angew. Chem. Int. Ed.* 2020, 19983; *Nat. Commun.* 2022, 327; *JACS* 2018, 11241; *Angew. Chem. Int. Ed.* 2020, 22763 for relevant insights). To further rule out the influence of SMSI, I suggest comparing the CO adsorption behavior under different conditions, as this would offer a clear distinction between the two mechanisms. The DFT results actually can not effectively support the mechanisms here.

Reply: Thank you for your comments. The XRD, HAADF-STEM imaging, and XPS results clearly indicate that the addition of trace amounts of sulfate species had almost no impact on the structure of Ru particles and TiO₂ support, therefore did not induce significant SMSI. Additionally, the H₂-DRIFTS and DFT calculation results suggest that the surface sulfate species at the interfacial Ru sites greatly enhanced the H₂ spillover process. Consequently, we confirm that the sulfur's influence on the selectivity change was not through the SMSI effect but by promoting hydrogen spillover. The operando FTIR data were acquired to elucidate the reaction process and provided supportive evidence on how sulfur influenced the selectivity. Therefore, we have included the H₂-DRIFTS and DFT calculation results in the main text, while the FTIR results are more appropriately placed in the Supplementary Information.

Your other concern is about how sulfur altered the catalytic pathways. Actually, we have already provided a comprehensive discussion in the Lines of 322-338 on Pages 18-19: "During the CO₂ hydrogenation on Ru/TiO₂ catalysts, the general reaction mechanism involved the initial adsorption of CO₂ at the Ru-TiO₂ interface, accompanied by H₂ activation and dissociation to H on the Ru sites. With the assistance of dissociated H, the adsorbed CO₂ could be activated to form intermediate CO. The presence of sufficient H and electrons allowed the intermediate CO to further convert into CH₄. We performed DRIFTS studies under steady-state CO₂ hydrogenation conditions, and it was observed that CO₂ was easily converted to intermediate CO on Ru/Ti-AR and Ru/Ti-S-AR at low temperature. Additionally, the intermediate CO adsorbed at Ru site of Ru/Ti-AR could be converted to CH₄ when the reaction temperature was above 473 K, while the intermediate CO adsorbed at Ru site of Ru/Ti-S-AR was stable and no CH₄ was obtained (Supplementary Fig. 18). In the case of Ru/Ti-S-AR, where the Ru-TiO₂ interface was modified by sulfate, the H transfer process was greatly enhanced. This led to more H and electrons migrating from Ru to TiO₂ via the S medium, resulting in fewer H atoms remaining on the Ru sites. Consequently, this could effectively result in the low product selectivity to CH₄. In contrast, on the sulfate-free Ru-TiO₂, the hydrogen spillover and charge transfer could not proceed effectively. Therefore, the hydrogenation of

adsorbed CO proceeded more smoothly, leading to the high CH₄ selectivity (Fig. 4d).” Most of your recommended literatures (*Angew. Chem. Int. Ed.* 2020, 19983; *Nat. Commun.* 2022, 327; *Angew. Chem. Int. Ed.* 2020, 22763) have already been mentioned and cited in the manuscript.

As evidenced by the experimental results, the influence of sulfur species on the selectivity was mainly through promoting the hydrogen spillover. Given that the impact of SMSI on the selectivity change has been ruled out, it may not be essential to conduct a comparative analysis of CO adsorption behavior under different conditions. Additionally, the DFT calculation results highlighted the significant enhancement of H₂ spillover by the surface sulfate species, already providing strong support for our conclusions.